

# Biotic and abiotic transformation of amino acids in cloud water: Experimental studies and atmospheric implications.

**Saly Jaber, Muriel Joly, Maxence Brissy, Martin Leremboure, Amina Khaled, Barbara Ervens, and Anne-Marie Delort\***

**Université Clermont Auvergne, CNRS, SIGMA Clermont, Institut de Chimie de Clermont-Ferrand, F-63000 Clermont-Ferrand, France**

\*Corresponding author: A-Marie.delort@uca.fr

**Abstract**

The interest for organic nitrogen and particularly for quantifying and studying the fate of amino acids (AA) has been growing in the atmospheric science community. However very little is

known about biotic and abiotic transformation mechanisms of amino acids in clouds.

In this work, we measured the biotransformation rates of 18 amino acids with four bacterial strains (*Pseudomonas graminis* PDD-13b-3, *Rhodococcus enclensis* PDD-23b-28, *Sphingomonas* PDD-32b-11 and *Pseudomonas syringae* PDD-32b-74) isolated from cloud water and representative of this environment. At the same time, we also determined the abiotic

(chemical, OH radical) transformation rates within the same solutions mimicking the composition of cloud water. We used a new approach by UPLC-HRMS to quantify free AA directly in the artificial cloud water medium without concentration and derivatization.

The experimentally-derived transformation rates were used to compare their relative importance under atmospheric conditions and compared to the chemical loss rates based on

kinetic data of amino acid oxidation in the aqueous phase. This analysis shows that previous estimates overestimated the abiotic degradation rates, and thus underestimated the lifetime of amino acids in the atmosphere as they only considered loss processes but did not take into account the potential transformation of amino acids into each other.

## 1. Introduction

The organic matter (OM) content of the cloud water phase is very complex; it has been described using Fourier-Transform Ion Cyclotron Resonance Mass Spectrometry (FT-ICR MS) (Bianco et al., 2018; Zhao et al., 2013) . These global analytical methods revealed a very large number of organic carbon, organic sulfur and organic nitrogen compounds. For instance, in

cloud water at the puy de Dôme, 5258 monoisotopic molecular formulas were assigned to CHO,



CHNO, CHSO, and CHNSO (Bianco et al., 2018). Organic nitrogen compounds contribute a significant fraction to the total nitrogen in cloud water (18%) (Hill et al., 2007) and in aerosol particles 7 – 10% in urban areas (Xu et al., 2017) or even exceed other nitrogen contributions in marine aerosol (Miyazaki et al., 2011). Among these organic nitrogen molecules, amino acids

(AA) have been recently analyzed and quantified in cloud droplets collected at the puy de Dôme station (Bianco et al., 2016a) and on the Cape Verde Islands (Triesch et al., 2020). AA were also quantified in rain collected in marine and sub-urban sites (Mace et al., 2003b, 2003a; Mopper and Zika, 1987; Sidle, 1967; Xu et al., 2019; Yan et al., 2015), and in fog samples in Northern California (Zhang and Anastasio, 2003). In cloud water, free AA concentrations range

from 2.4±2.0 to 74.3±43.8 µg C L$^{-1}$ at the rural site of the puy de Dôme (Bianco et al., 2016a) and from 17 to 757 µg C. L$^{-1}$ at the marine site of Cape Verde (Triesch et al., 2020). These AA are from biological origin and are building blocks of peptides (also called 'combined AA') and proteins. They are initially present in aerosols which are further dissolved in atmospheric waters (Matos et al., 2016). Primary and secondary atmospheric sources of AAs are discussed in

previous reviews (Cape et al., 2011; Sutton et al., 2011). Biomass burning (Zhu et al., 2020), grassland (Scheller, 2001), ocean (Triesch et al., 2020) and agricultural activities (Song et al., 2017) were identified as major emission sources of amino acids.

Although organic carbon has been studied for a long time by atmospheric scientists, the interest for organic nitrogen and particularly for quantifying and studying the fate of AAs has been

growing these last decades due to their specific properties. Some AA can act as ice nuclei, for instance L-leucine nucleates ice at -4.5°C (Szyrmer and Zawadzki, 1997). Their mass can also add to the hygroscopic fraction of cloud condensation nuclei due to their high water-solubility (Kristensson et al., 2010). Another point concerns the participation of AA in the global nitrogen and carbon cycles. For example, it has been estimated that organo-nitrogen compounds are a

significant fraction (28%) of the total nitrogen deposited (Zhang et al., 2012). Their ubiquity in living organisms makes their presence in atmospheric deposition very important for both terrestrial and aquatic ecosystems as AA represent the most bioavailable form of nitrogen (Cornell, 2011).

Finally, as part of the atmospheric OM, AA are expected to undergo chemical processes in the

atmospheric water phase (clouds, fog, aerosol). Due to their low volatility, it can be assumed that they are not present in the gas phase. However little is known on their transformation processes occurring in the atmospheric compartments, and particularly in clouds.

Concerning abiotic transformation (phototransformation and radical chemistry) in atmospheric waters, some studies determined kinetic rate constants (k) of AAs with radicals (e.g. OH)



(Scholes et al., 1965; Motohashi and Saito, 1993; Prütz and Vogel, 1976) , singlet oxygen ($^1O_2$) (Kraljić and Sharpatyi, 1978; Matheson and Lee, 1979; McGregor and Anastasio, 2001; Michaeli and Feitelson, 1994; Miskoski and García, 1993 ; McGregor and Anastasio, 2001) or ozone ($O_3$) (Ignatenko and Cherenkevich, 1985 ; Pryor et al., 1984). Based on such kinetic data, some studies have reported the time of life of amino acids in fog (McGregor and Anastasio, 2001) or in cloud water (Triesch et al., 2020). From these studies it is clear that some amino acids are transformed very rapidly, while others are almost never transformed within the time scale of fog or cloud life. When additional effect of $^1O_2$ was considered, MET, TRP, TYR and HIS remained the most degraded AA (McGregor and Anastasio, 2001). Among other mechanisms, this fast degradation could explain why these AA are usually among the less concentrated in aerosols (Barbaro et al., 2011, 2015; Helin et al., 2017; Mace et al., 2003a; Mashayekhy Rad et al., 2019; Matsumoto and Uematsu, 2005; Samy et al., 2013; Yang et al., 2004), in rain (Mace et al., 2003a; Xu et al., 2019; Yan et al., 2015) or in clouds (Triesch et al., 2020). The characterization of amino acids in dew showed differences depending on seasons, meteorological parameters and irradiation conditions (Scheller, 2001).

Even less is known about the abiotic transformation pathways of these amino acids, as only some AA have been studied in detail. Most mechanistic studies are limited to the transformation of AA (GLY, TRP, ASP, SER) into small carboxylic acids such as acetic, oxalic, malonic or formic acids (Berger et al., 1999; Bianco et al., 2016b; Marion et al., 2018). In some cases, an amino acid can be converted into another one or into very different molecules (Bianco et al., 2016b; Mudd et al., 1969; Prasse et al., 2018; Stadtman, 1993; Stadtman and Levine, 2003). The main concern with these mechanistic studies, is that they were performed under conditions rather far from atmospheric conditions. Incubation media did not contain a mixture of AA or real atmospheric samples. More they were sometimes measured with proteins in which the peptidyl bond might change the reactivity compared to free AA (Pattison et al., 2012).

Another missing aspect concerns the potential biotransformation of these AAs in atmospheric waters. The microbial community which is present in cloud waters is metabolically active (Amato et al., 2017, 2019; Vaïtilingom et al., 2012) and has been shown the biotransform mono and dicarboxylic acids, methanol, formaldehyde, phenol and catechol (Ariya et al., 2002; Husárová et al., 2011; Jaber et al., 2020; Vaïtilingom et al., 2010, 2011, 2013). It is well-known that microorganisms have enzymatic networks able to biodegrade or biosynthesize amino acids. These pathways are complex and very interconnected (KEGG pathway database) . However, no data exist about the biotransformation rates and metabolic pathways of AAs in cloud water.



The aim of the present study is thus to measure biotic and abiotic rates of transformation of free AA in microcosms mimicking cloud water with an incubation medium containing 19 AA, other major carbon (acetate, succinate, formate, oxalate) and nitrogen sources ($NH_4^+$, $NO_3^-$) as well as major salts (e.g., $Na^+$, $Cl^-$, $SO_4^{2-}$) present in cloud water collected at the puy de Dôme station

(Deguillaume et al., 2014). In addition, abiotic transformation rates are calculated based on rate constants of oxidation reactions with OH, $^1O_2$ and $O_3$ as reported in the literature. These experimental and theoretical rates of transformation are compared with each other and to previous literature studies and are discussed in terms of their atmospheric implications.

## 2.  Materials and Methods

### 2.1    Experiments in microcosms

The experiments of biotic and abiotic transformation of amino acids were performed in microcosms mimicking cloud conditions at the puy de Dôme station (1465 m). Solar light was fitted to that measured directly under cloudy conditions and the temperature (17°C) was representative of the average temperature in the summer. Incubations were performed in an

artificial cloud water medium containing inorganic ions, carboxylic acids and amino acids within the same range of concentrations as those measured in clouds that were impacted by marine air masses collected at the puy de Dôme station (Table S1, pH = 6.0) (Bianco et al., 2016a; Deguillaume et al., 2014). *Rhodococcus enclensis* PDD-23b-28, *Pseudomonas graminis* PDD-13b-3, *Pseudomonas syringae* PDD-32b-74 and *Sphingomonas* sp.PDD-32b-11 bacterial

strains were chosen because they belong to the most abundant and active bacterial genera in cloud water (Amato et al., 2017; Vaïtilingom et al., 2012). In addition, the complete genome sequences of *Rhodococcus enclensis* PDD-23b-28, *Pseudomonas graminis* PDD-13b-3, *Pseudomonas syringae* PDD-32b-74 have been published recently giving access to their metabolic pathways in more detail (Besaury et al., 2017a, 2017b; Lallement et al., 2017).

Bacterial cell concentrations used in the experiments (~$5\times10^5$ cells $mL^{-1}$) for biotransformation were consistent with those present in cloud water (Vaïtilingom et al., 2012). All experiments were performed in triplicates.

### 2.1.1   Cell preparation for further incubations

*Rhodococcus enclensis* PDD-23b-28, *Pseudomonas graminis* PDD-13b-3, *Pseudomonas*

*syringae* PDD-32b-74 and *Sphingomonas* sp. PDD-32b-11 were grown in 10 mL of R2A medium for 16 h at 17°C, 130 rpm (Reasoner and Geldreich, 1985). Then 1 mL of cultures were centrifuged at 12500 rpm for 3 min. Bacteria pellets were rinsed two times with 1 mL of





artificial marine cloud water, previously sterilized by filtration under sterile conditions using a 0.22 μm PES filter. The bacterial cell concentration was estimated by optical density at 600 nm using a spectrophotometer UV3100 to obtain a concentration close to $5 \times 10^5$ cell mL$^{-1}$. Finally, the concentration of cells was precisely determined by counting the colonies on R2A Petri dishes or by flow cytometry technique.

### 2.1.2 Biotransformation of amino acids

*Rhodococcus enclensis* PDD-23b-28, *Pseudomonas graminis* PDD-13b-3, *Pseudomonas syringae* PDD-32b-74 and *Sphingomonas* sp. PDD-32b-11 cells were each resuspended in a 50 mL flask of 1 μM amino acids (19 amino acids namely alanine (ALA, SIGMA), arginine (ARG SIMAFEX), asparagine (ASN, SIGMA), aspartate (ASP, Aldrich-Chemie), glutamine (GLN, SIGMA), glutamic acid (GLU), glycine (GLY, MERCK), histidine (HIS, SIGMA), isoleucine(ILE SIGMA-ALDRICH,), lysine(LYS, SIGMA-ALDRICH), methionine (MET, SIGMA), phenylalanine (PHE, ACROS organics), proline (PRO, SIGMA-ALDRICH), serine (SER, SIGMA,), threonine (THR, SIGMA), tryptophan (TRP, SIGMA), tyrosine (TYR, SIGMA-ALDRICH), valine (VAL, SIGMA-ALDRICH), cysteine (CYS, SIGMA-ALDRICH)/ 1 μM of each amino acid), prepared in artificial cloud water (Table S1) and incubated at 17 °C,130 rpm agitation for 7 hours in the dark. A control experiment was performed by incubating amino acids without bacteria; AA concentration remained stable over time (1 μM for each amino acid was obtained at the end of the experiment).

### 2.1.3 Abiotic transformation of amino acids

The same 19 amino acids, at a concentration of 1 μM each in the artificial cloud medium (Table S1) were incubated at 17°C, 130 rpm agitation for 7 hours in photo-bioreactors designed by Vaïtilingom et al. (2011).OH radicals were generated by photolysis adding 0.5 mM Fe-Ethylenediamine-*N,N'*-disuccinic acid (EDDS) complex solution. The Fe(EDDS) solution (iron complex with 1:1 stoichiometry) was prepared from iron(III) chloride hexahydrate (FeCl$_3$, 6H$_2$O; Sigma-Aldrich) and (S,S)-ethylenediamine-N,N'-disuccinic acid trisodium salt (EDDS, 35% in water). A complementary experiment was also performed consisting of incubation of this solution in the presence of light without Fe(EDDS) complex. The experimental conditions of the irradiation experiments (Sylvania Reptistar lamps; 15 W; 6500 K) and the mechanism of the •OH radical production under light irradiation are described by Jaber et al. (2020). Assuming steady-state conditions for •OH at the beginning of the experiments (i.e., equal •OH production and loss rates), an •OH concentration of $8.3 \cdot 10^{-13}$ M was calculated as described by



Jaber et al. (2020). This concentration is at the upper limit of $^{\bullet}$OH concentrations in cloud water as derived from various model studies (Arakaki et al., 2013; Lallement et al., 2018).

## 2.2 Analytical methods

### 2.2.1 Amino acid UPLC-HRMS Analyses

During the experiments in microcosms, 600 μL of the incubation medium were sampled regularly and centrifuged at 10 500 x g for 3 min and the supernatants were kept frozen until analyses. In order to quantify the amino acid concentrations in the incubations we developed here a new approach using a LC-HRMS technique based on a direct measurement by injection of the incubation medium without derivatization. All AAs could be quantified under these

conditions, except cysteine.

LC-HRMS analyzes of amino acids were performed using an UltiMate™ 3000 (Thermo Scientific™) UHPLC equipped with a Q Exactive™ Hybrid Quadrupole-Orbitrap™ Mass Spectrometer (Thermo Scientific™) ionization chamber. Chromatographic separation of the analytes was performed on BEH Amide/HILIC (1.7 μm, 100 mm x 2.1 mm) column with

column temperature of 30°C. The mobile phases consisted of 0.1% formic acid and water (A) and 0.1% formic acid and acetonitrile (B) with a flow rate 0.4 mL min$^{-1}$. A four-step linear gradient of 10% A and 90% B in 8 min, 42% A and 58% B in 0.1 min, 50% A and 50% B for 0.9 min, 10% A and 90% B for 3 min was used throughout the analysis.

The Q Exactive ion source was composed of an electrospray ionization (ESI+) and the Q-

Orbitrap™. Flow injection analyses were performed for individual amino acid solutions in order to obtain the mass spectra, from which ions were selected using the SIM (Selected Ion Monitoring) mode. The instrument was set for maximum ion throughput, the automatic gain control target or the number of ions to fill C-Trap was set to $10^5$ for a maximum injection time of 100 ms. Gas (N$_2$) flow rate and sheath gas (N$_2$) flow rate were set at 13 a.u. and 50 a.u.

respectively. Other parameters were as follows: 2 a.u for the sweep gas flow rate, 3.2 kV for the spray voltage in positive mode, 320°C and 425°C for the capillary temperature and the heater temperature, respectively. Under these conditions the mass resolution was 35000 fwhm. Analysis and visualization of the mass data were performed using Xcalibur™ 2.2 software (Thermo Scientific™).

Table S2 presents the retention times and values of m/z for the ions [M+H] measured under these conditions for each amino acid.



### 2.2.2 Calibration curves, LOD and LOQ determination

In order to quantify the amino acid concentrations, calibration curves were established for each experimental series of LC-HRMS analyses. In standard solutions, six concentrations of amino acids (0.01, 0.05, 0.1, 0.5, 1.0, 5.0 µM) were used for these external standard multipoint

calibrations. This range of concentrations is appropriate considering that the initial amino acid concentration in the biotic and abiotic transformation experiments is 1µM. Figure S1 presents an example of calibration curves for the 18 amino acids. The limits of detection (LOD) and quantification (LOQ) were calculated based on the standard deviation of the response (Sa) and on the slope of the calibration curves (b) (technical triplicate).

$$LOD = 3Sa/b \ µM$$

$$LOQ = 6Sa/b \ µM$$

The obtained values of LOD and LOQ were considered to be fit-for-purpose (Table S2).

### 2.2.3 Calculation of amino acids degradation rates in microcosms

The degradation rates of amino acids were calculated after normalization based on the ratio of

the concentration at time t ($C_t$) and the concentration at time t = 0 ($C_0$). The pseudo-first-order rate constants ($k_{isoleucine}$, $k_{valine}$ $k_{proline}$…) were determined using Equation 1:

$$Ln(C_t/C_0) = f(t) = - k_{amino \ acid} \ t \qquad [Eq-1]$$

The slopes at the origin were used to calculate the corresponding degradation rates. For biotransformation, the rates were corrected by the precise number of bacterial cells present in

the incubations and are expressed in the form of mol cell$^{-1}$ h$^{-1}$. An example is given in Figure S2a and b for the case of the biodegradation of GLN.

## 3. Results and discussion

### 3.1 Biotransformation of amino acids in microcosms

#### 3.1.1 Biotransformation rates of the 18 amino acids by the different bacterial strains

The biotransformation of alanine, arginine, asparagine, aspartate, glutamic acid, glycine, histidine, isoleucine, lysine, methionine, phenylalanine, proline, serine, threonine, tryptophan, tyrosine, valine and glutamine by four different bacterial strains isolated from cloud water at the puy de Dôme station in a marine artificial cloud medium was monitored in four independent microcosms containing only one of the strains. Figure 1 shows the results obtained for each

amino acid and each bacterial strain (*Rhodococcus enclensis* PDD-23b-28, *Pseudomonas graminis* PDD-13b-3, *Pseudomonas syringae* PDD-32b-74 and *Sphingomonas* sp. PDD-32b-



11). The standard error bars reflect significant biological variability measured from three triplicates (independent incubations). Note that the biotransformation rates of valine, isoleucine and glycine could be obtained only for one replicate due technical problems. Table 1 summarizes the average values of the biodegradation rates of the 18 amino acids for the four

bacterial strains. These average values for biodegradation (negative values) range from -1.03 $10^{-14}$ mol cell$^{-1}$ h$^{-1}$ to -8.0210$^{-17}$ mol cell$^{-1}$ h$^{-1}$, i.e. spanning a range of almost two orders of magnitude depending on the amino acid and the bacteria strain. Note that in the case of glycine and the strain *Pseudomonas graminis* PDD-13b-3, and of aspartate and the strain *Sphingomonas* sp. PDD-32b-11, the values are positive, indicating a net synthesis and not a net loss. The

incubations were performed in a complex medium containing all AA, and as a consequence the rate values are actually net values as all the AA are connected through metabolic pathways corresponding to both biodegradation and biosynthetic pathways (Figure S3).

Overall *Pseudomonas graminis* PDD-13b-3 appears to be the most active strain followed by *Rhodococcus enclensis* PDD-23b-28 (Figure 1, Table 1). However, for some amino acids, this

order is reversed, *Rhodococcus enclensi*s degrades alanine, asparagine, phenylalanine and tryptophan more efficiently than *P. graminis* does. For all amino acids, *Pseudomonas syringae* PDD-32b-74 is less active than *R. enclensis* and *P. graminis* followed by *Sphingomonas* sp. PDD-32b-1.1

Considering the best degrading strains (Figure 1 and Table 1), the most efficiently biodegraded

amino acids are in the order valine >alanine > arginine > glutamate > glutamine > lysine > proline > asparagine > arginine > serine > tryrosine > aspartate, with biodegradation rates within the range of $10^{-14}$ to $10^{-15}$ mol cell$^{-1}$ h$^{-1}$. A second group of AA have lower biodegradation rates in the range of $10^{-16}$ to $10^{-17}$ mol cell$^{-1}$ h$^{-1}$ in the following order: phenylalanine > threonine > histidine > methionine > glycine >isoleucine > tryptophan.

### 3.1.2 Link of the biodegradation rates with metabolic pathways

In bacteria many amino acids are connected within the same metabolic pathways *via* the enzymatic activities of their biosynthesis or biodegradation. Figure S3 presents a simplified network of the AA metabolic pathways as described in KEGG pathway database where the AA

belonging to the same pathway are shown in the same color. We investigated the hypothesis of a potential link between the rates of biodegradation for each amino acid by the four strains with their connection in specific metabolic pathways (Figures S3 and S4). Glutamate, glutamine, proline and arginine metabolic pathways are closely linked (blue boxes in Figure S3) and in parallel their biodegradation rates are on the same order of magnitude (Figure S4). This is also





true for the group of serine, threonine glycine and methionine (yellow boxes in Figure S3), and for the group tyrosine, phenylalanine and tryptophan (green boxes in Figure S3), respectively. Alanine, asparagine and aspartate (purple boxes in Figure S3) are also related in the network, although the rate of biodegradation of aspartate is lower compared to the other two. Valine and

isoleucine biodegradation rates are quite different; this can be explained by two divergent routes: valine is produced from pyruvate, while isoleucine is formed from 2-oxobutanoate. Histidine has a unique metabolic pathway, while lysine is also a special case as two metabolic routes exist: one is linked to 2-oxoadipate, the other is connected to alanine, aspartate and asparagine. To conclude, the rates of biodegradation can be grouped according to their presence

in common metabolic pathways. This could explain, as suggested by (Scheller, 2001), why in dew, the concentrations ARG, PRO and GLU, three AA belonging to the same pathway and connected to the urea cycle (Figure S3), were increasing simultaneously.

### 3.1.3 Dependence of the selectivity of AA biodegradation on the bacterial phylogeny

The rates of biodegradation of the different amino acids expressed as a percentage of the highest

rate for each strain are presented in the form of a radar plot in Figure S5. A clear difference is observed between *Rhodococcus enclensis* PDD-23b-28 belonging to Actinobateria (Figure S5a) and the other strains belonging to Proteobacteria (Figure S5b and c). Within Proteobacteria, it is possible to distinguish *Sphingomonas* sp PDD-32b-11 (Figure S5b) belonging to α-Proteobacteria from *Pseudomonas graminis* PDD-13b-3 (grey, Figure S5c) and *Pseudomonas*

*syring*ae PDD-32b-74 (yellow, Figure S5c) belonging to γ-Proteobacteria. In addition, the two *Pseudomona*s strains share very similar trends. So, although the biodegradation rates of *P. syringae* are much lower than those of *P. graminis,* they seem to transform preferentially the same type of amino acids. This should be confirmed with a larger set of isolates. It suggests that the selectivity of AA biodegradation could be related to the phylogeny of the bacterial strains.

### 3.2     Abiotic transformation of amino acids in microcosms

The abiotic transformation rates of the amino acids measured in the experiments in our microcosms are shown in Table 2 and Figure 2. The first important result is that some amino acids are degraded (TYR, THR, MET, TRP, SER, GLU, VAL, HIS, ALA, ILE) while others are produced (ASN, PRO, GLY, ARG, LYS, GLN, ASP). Abiotic degradation rates (negative

values of the transformation rates) were within the range of $-7.98 \ 10^{-8}$ to $-9.70 \ 10^{-7} \ \text{mol h}^{-1} \ \text{L}^{-1}$. Net abiotic production rates (positive values) were within the range of $7.69 \ 10^{-8}$ to $1.05 \ 10^{-6}$ $\text{mol h}^{-1} \ \text{L}^{-1}$, except for ASP whose rate was very high $(3.79 \ 10^{-5} \ \text{mol h}^{-1} \ \text{L}^{-1})$. As mentioned in the context of biotic transformations (Section 3.1.1), the incubations are performed in artificial





cloud media containing the mixture of the 19 AA, and, thus, the measured rates of abiotic transformations are net values, integrating various mechanisms.

### 3.3 Comparison of amino acid biotic and abiotic transformation rates

### 3.3.1 Kinetic rate constants for chemical oxidation reactions

In order to assess the atmospheric importance for the transformation of individual amino acids, we make the following assumptions. Loss by OH reactions occur with the rate constants listed in Table S3 and an OH(aq) concentration of $1 \cdot 10^{-14}$ M (Arakaki et al., 2013). For the oxidation by ozone, ozone has a concentration in cloud water of 0.5 nM which corresponds to a gas phase

mixing ratio of 50 ppb, using $K_H(O_3) \sim 10^{-3}$ M atm$^{-1}$ (Sander, 2015). It has been shown previously that the rate constants of amino acids with ozone are strongly pH dependent, with smaller values for the protonated amino form (McGregor and Anastasio, 2001). Since the first acid dissociation constants (pKa$_1$) for all amino acids are in the range of 2 – 2.5 and the second acid dissociation constants (pKa$_2$) (de/protonation of the amino group) in the range of 9 – 9.5

(Haynes, 2010), it can be assumed that at cloud-relevant pH values (3 < pH < 6) the amino acids are present as carboxylates with protonated amine groups. In addition, we also consider the oxidation by singlet oxygen $^1O_2$. Kinetic rate constants for only about half of the amino acids are available (Table S3). The estimates for $^1O_2$ concentrations in the atmospheric aqueous phase are much sparser and less constrained than for the other oxidants. However, several studies

agree that its concentration may be two to three orders of magnitude higher than the OH radical in clouds, fogs and aerosol particles, respectively (Faust and Allen, 1992; Manfrin et al., 2019). Therefore, we assume an aqueous concentration of $[^1O_2(aq)] = 10^{-12}$ M here. Other oxidants (e.g. HO$_2$/O$_2^-$, NO$_3$) are not included in our analysis as based on the few available kinetic data, it can be estimated that reaction rates may be too slow to represent an efficient sink (McGregor

and Anastasio, 2001).

### 3.3.2 Comparison of biotic and abiotic transformation rates

In order to compare the relative importance of biotic (microbial) and abiotic (chemical) transformations under atmospheric conditions, we weight the experimentally derived biotransformation rates by the relative abundance of the various bacteria strains as found in

cloud water. An average concentration of $6.8 \cdot 10^7$ bacterial cells per liter of cloud water was identified in cloud water samples at the puy de Dôme (France) (Vaïtilingom et al., 2012). Further characterization of these samples showed that Actinobacteria (*Rhodococcus enclensis*



PDD-23b-28), α-Proteobacteria (*Sphingomona*s sp PDD-32b-1) and γ-Proteobacteria (*Pseudomonas graminis PDD-13b-3 and Pseudomonas syringae PDD-32b-74*) contributed to 6.3%, 16.2% and 29.8%, respectively, to the total cell concentration (Amato et al., 2017); the remaining 47.7% belonged to other phyla or classes (Bacteroidetes, beta-Proteobacteria, Firmicutes…).

Using these relative contributions, the loss rates as observed in our experiments (Section 3.1 and 3.2) were used to compare the loss rates under atmospheric conditions. For this comparison, we calculated the biotransformation rates in cloud water as

$$\frac{d[AA]}{dt} = -0.063 \, k_{23b2?} \cdot 1.91 - 0.162 \, k_{3?} \quad \cdot 1.91 - \frac{29.8}{2} k_{23b28b} \cdot 1.91 - \frac{29.8}{2} k_{13b2} \cdot 1.91 \qquad \text{[Eq-2]}$$

We scaled each contribution by a factor 1.91 (= 100/52.3) implying that the four bacteria types are representative for the remainder (47.7%) of the bacteria population.

We compare these rates to the photochemical rates derived in the experiments (Section 3.2).

15    However, since the experiments where conducted with OH concentrations likely higher than ambient ones in cloud water, we correct these rates to OH(aq) concentrations in clouds by

$$\left(\frac{d[AA]}{dt}\right)_{phot?,exp} = -k_{photo,exp} \cdot \frac{[OH(aq)]_{ph? .exp}}{[OH(aq)]_{cloud}} \qquad \text{[Eq-3]}$$

20    with $[OH(aq)]_{photo,exp} = 8.3 \cdot 10^{-13}$ M and $[OH(aq)]_{cloud} = 1 \cdot 10^{-14}$ M.

Finally, these experimentally-based abiotic transformation rates based on the experiments are compared to those calculated based on kinetic data only.

$$\left(\frac{d[AA]}{dt}\right)_{cloud} = -k_{OH}[? (? )]_{cloud} - k_{O3}[O_3(? )]_{cloud} - k_{1O2}[ {}^1O_2(? )]_{cloud}$$

[Eq-4]

In previous studies, the reactivity towards the OH radical and/or other oxidants was compared in terms of half-lives τ. However, we chose not to present half-lives here because net production terms as observed in the experiments cannot be represented and would result in unphysical, negative values for τ.

30    The comparison between the rates calculated by Equations 2 - 4 is shown in Figure 3. For some of the acids (ALA, GLU, THR) the predicted losses by OH from both approaches (photochemical experiments and based on kinetic data) are similar. For some AAs, it is predicted that the oxidation by ozone might contribute significantly more to their loss. For





several of the acids, biotransformation is predicted to exceed the loss by chemical reactions (e.g. ALA, ASN, GLU, PRO, VAL), for the bacteria cell and oxidant concentrations considered here. Note that the loss rates calculated by Equation 4 cannot reproduce the observed production of the various acids as observed in the experiments with the mixture of all amino acids.

### 3.3.3   Amino acid conversions

The oxidation of amino acids by a variety of oxidants has been performed in lab experiments. Results of such experiments are summarized in Table S4. It is obvious that generally most oxidation reactions lead to smaller fragmentation products and not to amino acids, independent

of the oxidant. A detailed discussion of the previously suggested reaction mechanisms of OH and/or $HO_2/O_2^-$ initiated amino acid oxidation has been given by Stadtman and Levine (2003). The studies summarized in Table S4 were not motivated by the investigation of amino acid oxidation pathways in the atmospheric aqueous phase. However, our experimental results suggest that some of the amino acids may be the product of oxidation reactions from precursor

amino acids, in qualitative agreement with some of the experiments listed in Table S4. The products and their distributions, however, are different than in the metabolic pathways shown in the KEGG mechanism (Figure S3). There are some similarities between the biotransformation and oxidation products, such as the formation of aspartic acid and asparagine from histidine, tyrosine formation from phenylalanine and glutamic acid formation from

proline. However, as the yields in the oxidation reactions were not reported, the efficiency of the various pathways for the formation of these acids cannot be estimated.

Our experiments suggest that amino acids can not only be chemically degraded in cloud water but also produced. While such transformation cycles are known from biological systems (KEGG mechanism, Figure S3), the production of amino acids by oxidation reactions in cloud

water has not been discussed in the literature. Previous model studies of amino acids in the atmospheric aqueous phase only compared the half-life times of the acids to each other or for different oxidants, solely based on kinetic data (McGregor and Anastasio, 2001; Triesch et al., 2020). Our study suggests that such estimates underestimate the concentrations of amino acids in the atmosphere since they ignore any production. These findings are qualitative as the product

yields and distributions are not known. Many of the experiments listed in Table S4 were performed under conditions that are not necessarily atmospherically relevant.





## 4. Summary, conclusions and atmospheric implications

We measured the biotic (microbial) transformation rates of 18 amino acids with four bacteria strains (*Pseudomonas graminis* PDD-13b-3, *Rhodococcus enclensis* PDD-23b-28, *Sphingomonas* sp. PDD-32b-11 and *Pseudomonas syringae* PDD-32b-74) that have been previously identified as being representative of the microbial communities in cloud water. At the same time, we also determined the abiotic (chemical, OH radical) transformation rates within the same solutions that resembled the composition of cloud water. We used a new approach by UPLC-HRMS to quantify free AA directly in the artificial cloud water medium without concentration and derivatization, improving the technique used in cloud water by (Triesch et al., 2020). This direct MS method avoids time-consuming and potential biases.

We used our experimentally-derived transformation rates to compare their relative importance under atmospheric conditions, i.e., for atmospherically relevant bacteria cell and OH concentrations in cloud water. These rates were compared to the chemical loss rates based on kinetic data of oxidation reactions of amino acids in the aqueous phase, as they were used previously to derive lifetimes of amino acids in the atmosphere. Our experiments show that previous estimates overestimated the degradation rates, and thus underestimated the lifetime of amino acids in the atmosphere as they only considered kinetic data describing loss processes but did not take into account the transformation of amino acids into each other. While such transformation cycles are well known for metabolic pathways (KEGG pathways), the mechanisms for chemical transformations are poorly constrained.

Our study qualitatively suggests that the sources and distribution of amino acids in the atmospheric particle and aqueous phases can be modified by metabolic and chemical transformation pathways. The distribution and abundance of specific amino acids in particles has been used in previous studies to conclude on aerosol sources (Barbaro et al., 2014, 2015). However, efficient abiotic and or biotic amino acid transformations during aerosol transport might alter the distribution and concentrations of amino acids so that source contributions might be more complex.

Free amino acids can represent up to 5% of WSOC in submicron sized particles but only 0.04% of WSOC in supermicron sized particles (Triesch et al., 2020) , or 9.1% of DOC in cloud water (Bianco et al., 2016a). Free AA can also represent 0.4% and 0.05% of WSON in submicron and supermicron sized particles (Triesch et al., 2020). Total hydrolysed AA (THAA) can account for 0.7 to 1.8% of DOC and from 3.8 to 6.0% of DON in rain samples (Yan et al., 2015) and from 6.2 to 23 % of DOC in fog sample (Zhang and Anastasio, 2003). Considering that WSON contributes to 25% of TDN of ambient aerosols (Lesworth et al., 2010) and WSOC contributes





to 20% of TOC (Saxena and Hildemann, 1996), the understanding of the lifetime and transformation rates of amino acids are essential, in order to characterize their atmospheric abundance and residence time. Our study highlights the need for further mechanistic investigations of the biotic (metabolic) and abiotic (chemical) transformations of amino acids

5   under conditions relevant for the atmospheric aqueous phases (clouds, fogs, aerosols). Such data should be used in atmospheric multiphase models to explore the role and competition of biotic and abiotic processes for the transformation and loss of amino acids and related compounds.



**Data availability:** All data can be obtained from the authors upon request.

**Ethics statements:** This work does not involve human or animal subject. There is no ethical problem.

**Author contributions:** AMD designed the experiments in microcosms. SJ, MB, MJ performed the experiments. BE and AK made the calculations for the abiotic and biotic transformations. AMD, MJ and BE wrote the manuscript. All the authors read and corrected the manuscript.

**Competing interests**: The authors declare that they have no conflict of interest.

**Acknowledgements:** This work was funded by the French National Research Agency (ANR) in the framework of the 'Investment for the Future' program, ANR-17-MPGA-0013. S. Jaber is recipient of a school grant from the Walid Joumblatt Foundation for University Studies (WJF), Beirut, Lebanon and M. Brissy from Clermont Auvergne Metroplole. The authors also thank the I-Site CAP 20-25.

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



## Figure captions

**Figure 1:** Biotransformation rates obtained for each amino acid and each bacterial strain (*Pseudomonas graminis* PDD-13b-3 (black), *Rhodococcus enclensis* PDD-23b-28 (blue),
5  *Sphingomonas* sp. PDD-32b-11 (red) and *Pseudomonas syringae* PDD-32b-74 (orange). The experiments were performed in microcosms containing the mixture of the 19 AA in a cloud artificial medium. The standard error bars reflect the significant biological variability measured from 3 triplicates (independent incubations).

10  **Figure 2:** Abiotic transformation rates $(\text{mol}\,\text{h}^{-1}\,\text{L}^{-1})$ obtained for each amino acid in microcosms containing the mixture of the 19 AA in a cloud artificial medium under irradiation in the presence of Fe(EDDSS) as source of OH radicals. The standard error bars reflect the variability measured from 3 triplicates (independent experiments). Negative values represent abiotic degradation while positive values represent abiotic production.

**Figure 3**: Reaction rates for 18 amino acids as observed based on experiments in the present study, scaled to atmospheric conditions (Eqs. 2 and 3) and rates for loss reactions by OH(aq), $O_3$(aq) and $^1O_2$(aq) (Eq-4)



Figure 1







Figure 2

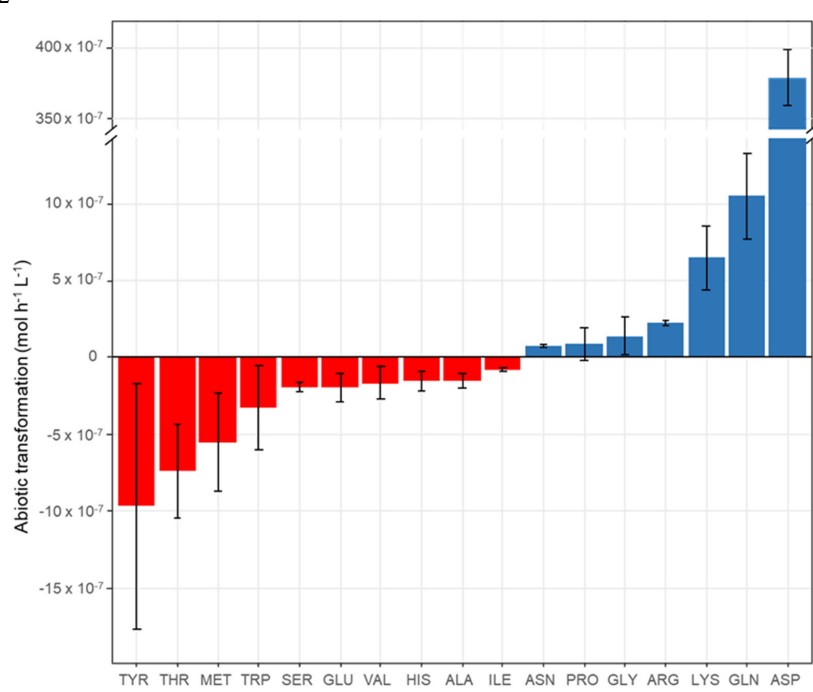





Figure 3

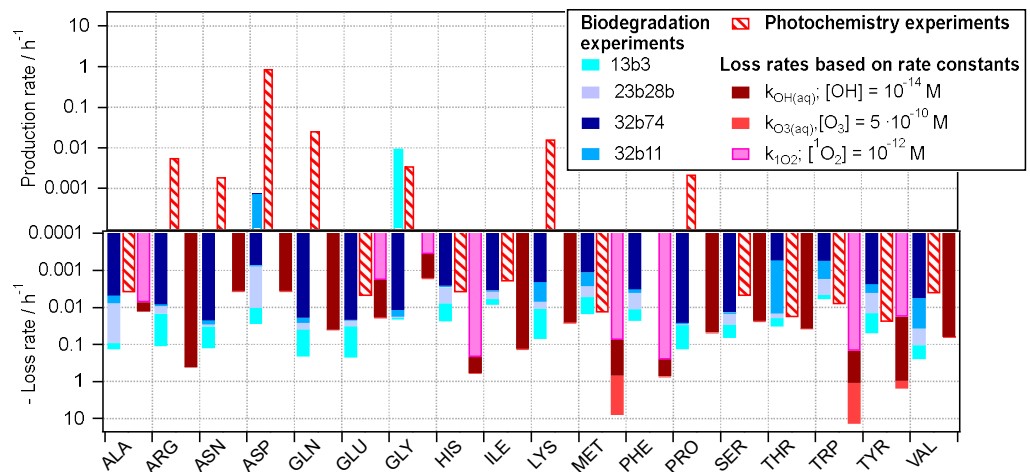



**Table 1**: Average values of the biotransformation rates (mol bact$^{-1}$ h$^{-1}$) of 18 amino acids by the four bacterial strains (*Pseudomonas graminis* PDD-13b-3, *Rhodococcus enclensis* PDD-23b-28, *Pseudomonas syringae* PDD-32b-74 and *Sphingomonas* sp. PDD-32b-11) and by the combination of these strains as representative of the biodiversity in a real cloud (named "Cloud") as described in section 3.3.2.

Positive values correspond to a net biosynthesis, while negative ones correspond to a net biodegradation.

| | VAL | ALA | GLU | GLN | LYS | PRO | ASN | ARG | SER |
|---|---|---|---|---|---|---|---|---|---|
| **13b-3 *Pseudomonas graminis*** | -7.29 x 10$^{-15}$ | -2.19 x 10$^{-15}$ | -9.89 x 10$^{-15}$ | -8.72 x 10$^{-15}$ | -3.05 x 10$^{-15}$ | -5.29 x 10$^{-15}$ | -4.67 x 10$^{-15}$ | -4.99 x 10$^{-15}$ | -1.90 x 10$^{-15}$ |
| **23b-28 *Rhodococcus enclensis*** | -8.11 x 10$^{-15}$ | -1.03 x 10$^{-14}$ | -1.27 x 10$^{-15}$ | -1.62 x 10$^{-15}$ | -5.07 x 10$^{-16}$ | -2.33 x 10$^{-16}$ | -6.14 x 10$^{-16}$ | -7.03 x 10$^{-16}$ | -1.81 x 10$^{-15}$ |
| **32b-11 *Sphingomonas* sp.** | -1.48 x 10$^{-15}$ | -1.46 x 10$^{-16}$ | -3.91 x 10$^{-17}$ | -3.22 x 10$^{-16}$ | -2.31 x 10$^{-16}$ | -1.08 x 10$^{-16}$ | -2.89 x 10$^{-16}$ | -3.98 x 10$^{-17}$ | -6.82 x 10$^{-17}$ |
| **32b74 *Pseudomonas syringae*** | -2.91 x 10$^{-16}$ | -2.40 x 10$^{-16}$ | -1.09 x 10$^{-15}$ | -9.87 x 10$^{-16}$ | -1.05 x 10$^{-16}$ | -1.33 x 10$^{-15}$ | -1.14 x 10$^{-15}$ | -4.21 x 10$^{-16}$ | -6.88 x 10$^{-16}$ |
| **Cloud** | -3.60 x 10$^{-15}$ | -1.97 x 10$^{-15}$ | -3.29 x 10$^{-15}$ | -3.06 x 10$^{-15}$ | -1.03 x 10$^{-15}$ | -1.95 x 10$^{-15}$ | -1.82 x 10$^{-15}$ | -1.64 x 10$^{-15}$ | -9.75 x 10$^{-16}$ |

| | TYR | THR | ASP | HIS | PHE | MET | GLY | ILE | TRP |
|---|---|---|---|---|---|---|---|---|---|
| **13b-3 *Pseudomonas graminis*** | -1.79 x 10$^{-15}$ | -6.83 x 10$^{-16}$ | -9.07 x 10$^{-16}$ | -8.08 x 10$^{-16}$ | -6.19 x 10$^{-16}$ | -5.14 x 10$^{-16}$ | 4.85 x 10$^{-16}$ | -1.35 x 10$^{-16}$ | -7.48 x 10$^{-17}$ |
| **23b-28 *Rhodococcus enclensis*** | -1.23 x 10$^{-15}$ | -5.53 x 10$^{-16}$ | -1.14 x 10$^{-15}$ | -5.91 x 10$^{-16}$ | -8.57 x 10$^{-16}$ | -3.12 x 10$^{-16}$ | -1.96 x 10$^{-16}$ | -2.78 x 10$^{-16}$ | -3.45 x 10$^{-16}$ |
| **32b-11 *Sphingomonas* sp.** | -8.02 x 10$^{-17}$ | -6.76 x 10$^{-16}$ | 3.41 x 10$^{-17}$ | -1.42 x 10$^{-17}$ | -4.42 x 10$^{-17}$ | -7.40 x 10$^{-17}$ | -2.89 x 10$^{-16}$ | -7.50 x 10$^{-18}$ | -5.50 x 10$^{-17}$ |
| **32b74 *Pseudomonas syringae*** | -1.19 x 10$^{-16}$ | -2.71 x 10$^{-17}$ | -3.69 x 10$^{-17}$ | -1.31 x 10$^{-16}$ | -1.64 x 10$^{-16}$ | -5.78 x 10$^{-17}$ | -6.01 x 10$^{-16}$ | -1.78 x 10$^{-16}$ | -2.85 x 10$^{-17}$ |
| **Cloud** | -7.16 x 10$^{-16}$ | -4.78 x 10$^{-16}$ | -3.96 x 10$^{-16}$ | -3.43 x 10$^{-16}$ | -3.40 x 10$^{-16}$ | -2.23 x 10$^{-16}$ | -1.46 x 10$^{-16}$ | -1.25 x 10$^{-16}$ | -8.80 x 10$^{-17}$ |





**Table 2**: Abiotic transformation rates of the 17 AA (mole $h^{-1}$ $L^{-1}$) measured in microcosms containing the mixture of the all AAs in a cloud artificial medium under irradiation in the presence of Fe(EDDSS) as source of OH radicals. Positive values represent degradation while negative values represent production. Mean values were calculated from 3 triplicates (independent experiments) except for ASN, GLN, GLY and PRO. No value could be obtained for PHE (technical problem).

| | TYR | THR | MET | TRP | SER | GLU | VAL | HIS | ALA | ILE |
|---|---|---|---|---|---|---|---|---|---|---|
| **Degradation** | -9.70 $\times 10^{-7}$ | -7.41 $\times 10^{-7}$ | -5.55 $\times 10^{-7}$ | -3.29 $\times 10^{-7}$ | -1.97 $\times 10^{-7}$ | -1.96 $\times 10^{-7}$ | -1.67 $\times 10^{-7}$ | -1.53 $\times 10^{-7}$ | -1.53 $\times 10^{-7}$ | -7.98 $\times 10^{-8}$ |

| | ASN | PRO | GLY | ARG | LYS | GLN* | ASP |
|---|---|---|---|---|---|---|---|
| **Production** | 7.69 $\times 10^{-8}$ | 8.82 $\times 10^{-8}$ | 1.40 $\times 10^{-7}$ | 2.25 $\times 10^{-7}$ | 6.47 $\times 10^{-7}$ | 1.05 $\times 10^{-6}$ | 3.79 $\times 10^{-5}$ |