# Peer review of "Biotic and abiotic transformation of amino acids in cloud water: Experimental studies and atmospheric implications."

_Biogeosciences, 2020_

## Referee Comment (RC1) · Anonymous Referee #1 · 24 Aug 2020

The authors present a very interesting work, measuring biotic and abiotic transformation rates of amino acids under cloud water conditions. The topic is very relevant, the approach is innovative and the results are promising. The manuscript is written in an understandable way and reads very well. Some improvements on the Figures are needed. This work is suitable for the journal; however, some comments and questions should be addressed.

I have some questions and comments about the analytical method: A concentration of 1 $\mu$mol of each amino acid was applied for the experiments. How does this concentration compare to ambient amino acid concentrations? And, even more important:

[Figure]

how are typical compositions of amino acids in the ambient atmosphere? Is a uniform concentration of 1 $\mu$mol for each amino acid realistic? This might strongly influence the different degradation pathways. Please comment on that and I'd recommend to include such discussions in the manuscript.

Concerning the analytical method; the authors used ESI. It is known that ESI is prone the matrix effects (ion suppression) especially in ambient samples containing salt. Therefore, a sample preparation method is often applied, to eliminate disturbing matrix compounds. Did the authors test such effects, as ion suppression for the individual amino acids, for example by comparison of the external calibration to standard addition?

The LOQs seem quite high. How do they compare to other analytical methods used for amino acid analytics? It seems that the LOQ are close to the applied concentration of 1 $\mu$mol, so did this cause problems in the analytical accuracy? How was the precision (e.g. standard deviation) of the analytical method? As the authors introduce the analytical method as a new approach and an improved technique, some further method validation would be necessary in my opinion.

How about contaminations? Did you measure blanks and if so, were they considered? Finally; would you analytical method (without pre-concentration and sample preparation) be applicable for measuring amino acids in ambient marine samples?

Chapter 2.1.: The authors explained that the strains were chosen because they are the most abundant and active bacteria in cloud water. Are there more information on these strains available, that might be used to explain their different behaviour towards the individual amino acids?

Chapter 3.1.1: Interestingly, the efficiencies of the different strains are very variable among each other and concerning the different amino acids. The authors mentioned that all amino acids were mixed together in the experiment. I was wondering if you also performed these experiments with single amino acids? This might be interesting

especially regarding the net production of GLY that is certainly a product from the degradation of other amino acids.

Chapter 3.1.2.: The manuscript often refers to the Figures S3 and S4 which seem to be crucial for following and understanding the text. As the manuscript does not contain many Figures, maybe transfer them to the main part? An alternative could be to highlight the amino acids that have the same metabolic pathway in Figure 1 (instead of Figure S4). The statement that the "blue box" amino acids exhibit the same behaviour regarding their biodegradation is difficult to see in Figure S4 and a "zoom in" would be required. Actually, it seems that GLY shows quite a different behaviour, not in line with the other "blue box" amino acids. Also the "green box" amino acids are difficult to see (Fig. S4). For the "purple box" amino acids; the mentioned strong similarities are not obvious from Fig. S4. The 23b-28 strain seems to be much stronger for ASN compared to ALA. Please re-think the way of showing the similarities and maybe find a clearer way to present similarities and differences for the metabolomic-groups amino acids and their response to the different strains.

Chapter 3.1.3 Are there any more detailed explanation theories why these different strains exhibit such different behaviours? To what properties could that be related? On page 9, line 24 the authors mention that the AA biodegradation could be linked to the phylogeny of the bacterial strains. Could you give some more explanation (to non-biologists) about this?

Chapter 3.3.1: I wonder how relevant singlet oxygen is for diluted systems. (lifetime?) Is the sink for singlet oxygen considered in the rates (Fig 3)?

Figure 2 shows that degradation and formation happens for the individual amino acids. As a general question and also related to Fig. 2: Can any mechanisms for the formation/degradation of the individual amino acids be derived from that?

Chapter 3.3.2 and Figure 4: This chapter deals with the comparison of the biotic and the abiotic pathway. They are shown in Fig.3. While some exemplary comparisons are

made between both pathways (Page 11, Line 30 - Page 12 Line 6) I miss some real conclusions here. In addition, Figure 3 is difficult to understand and not well discussed; some more details might be helpful to understand the outcome of Fig. 3.

Chapter 4: The conclusions are well written. The authors summarize that the so far only degradation (losses) of amino acids but not production (transformation into each other) was considered. However, I was struggling with the following sentence: "Our study qualitatively suggests that the sources and distribution of amino acids in the atmospheric particle and aqueous phases can be modified by metabolic and chemical transformation pathways." -> Could the authors derive more precise conclusions here? I understood it was the aim to show HOW the two pathways (biotic, abiotic) contribute. I was wondering if the authors could finally comment on the relative importance of the biotic and the abiotic pathway e.g. which seems to be the more important way?

Small comments: There are several typos e.g. page 2 line 11 (C.L-1), sometimes the chemicals / amino acids are written with capital letter, sometimes with small letters (e.g. Table S4).

Empty spaces are missing and the formulas in eq. 2-4 are not represented right. In addition, the reference style needs revisions (e.g. page 16, line 44-45, page 17, line 11, page 19, line 25.

Table S3: There are missing references (for GLU, GLY, SER. . .). At what temperature was the rate constant obtained?

Concerning the Data availability I'd strongly recommend to upload the data in a public database such as PANGAEA or similar.

Author contributions: I was surprised that "SJ", as the first author, did not "write the manuscript"?

―――――――――――――――――――

---

## Referee Comment (RC2) · Anonymous Referee #2 · 25 Aug 2020

In this work the authors conducted microcosm experiments with the aim of differentiating the roles of biotic and abiotic transformation of free amino acids in cloud water. In the experiment, they utilized 19 types of amino acids, four bacterial strains or photo-bioreactors, attempting to mimic ambient cloud conditions. With the kinetic loss/production data of amino acids, they concluded that previous studies may have overestimated the abiotic degradation rates of amino acids, and future modeling efforts should take the biotic and abiotic transformation of amino acids into account. Overall, I think the authors did a solid job in terms of writing and offered an interesting dataset in amino acid dynamics in the field of cloud chemistry. The data seem to be sound. However, I have some concerns the authors need to address before publication.

The authors claimed that they developed a new analytical technique that can analyze amino acids, without the need of any preparation such as derivatization, using UPLC-HRMS. This is nice effort, but I am surprised that the incubation medium they used, containing mM of ions such as Na, Ca an K etc. (Table S1), which would directly impact ion source and ionization process, was directly injected to the LC-MS. I don't think any mass spectrometry expert would be happy if you inject such a solution to the instrument. I am wondering what's the inject volume they used and how they can maintain a consistent sensitivity with such high ion strength solution (or how long). In addition, external calibration curves were used to quantify the amino acids in cloud water medium. Did you use the same water medium for the standards? If not, this might be a problem with the matrix effect. It may be also helpful to show the LC-MS chromatograms in the supplementary section. The bottom line is that more info is needed for this new approach you developed.

I like the experimental approach, such as clearly separating abiotic and biotic factors, and using free amino acids and single bacterial strains, which allowed you to tease out the convoluted factors observed in field samples. However, the authors need to realize/justify their experimental conditions which I think are far away from those of the field, thus more discussion is needed. For example, they used 1 uM of 19 types of amino acids, which represent 19uM amino acids or 684 ug C/L (assuming 3 C per amino acid); in contrast, the cloud water only contained 2.4-74.3 ug C/L, cited from the Introduction of the manuscript. Your rate calculation is dependent on the concentration, thus the extremely high concentrations you used could have led to a conclusion not relevant to the field (the rate constant could also change depend on how the bacteria take up the substrate). In addition, when bacteria are just harvested from culture medium, in a way they have been pre-trained to take up labile organic matter rapidly, thus the biotic loss rate you obtained could have been overestimated. Bacteria in the cloud water, on the contrary, may not be that active often due to substrate limitation. Similarly, the abiotic transformation rates would be different when you have an organic matrix present, like cloud water. As mentioned in the Introduction, organic matter in the cloud water is

complicated, including many different compounds, which may include quenchers and photosensitizers, the rates you obtained may not represent those of field. I think all these need to be factored in when you try to argue against previous studies, or apply these data to the field. I would like to see more discussion along these perspectives.

Many places in the Introduction, there are too many references which kind of stops the flow. I would suggest you only choose the key ones.

Page 3 line 28: "biotranform"? You meant: "...shown to biotransform..." Page 7 line 25: no need to list all these amino acids here. Page 8 line 3: should be "due to" Page 8 line 7: should be "bacterial strain" Page 8 line 9: I think "production" is a better word than "synthesis" here. Page 9 line 26: delete "in the experiments", redundant Page 11 line 5: delete the "..." Page 11 lines 10&17: the equations did not show up right. It might have something to do with the formatting. Page 12 line 8: delete "It is obvious that" Page 12 line 30: right, but as I mentioned before, your experimental conditions may not be that relevant, either.

---

## Author Comment (AC1) · 1 Oct 2020

**Answers to reviewer #1**

Referee Comment:

The authors present a very interesting work, measuring biotic and abiotic transformation rates of amino acids under cloud water conditions. The topic is very relevant, the approach is innovative and the results are promising. The manuscript is written in an understandable way and reads very well. Some improvements on the Figures are needed. This work is suitable for the journal; however, some comments and questions should be addressed.

Authors' Response:

We thank the reviewer for the very positive evaluation of our work and their constructive comments. We address all points in detail below.

Referee Comment:

I have some questions and comments about the analytical method: A concentration of 1 µmol of each amino acid was applied for the experiments. How does this concentration compare to ambient amino acid concentrations? And, even more important: how are typical compositions of amino acids in the ambient atmosphere? Is a uniform concentration of 1µmol for each amino acid realistic? This might strongly influence the different degradation pathways. Please comment on that and I'd recommend to include such discussions in the manuscript.

Authors' Response:

The amino acid (AA) concentrations and their ratios to each other in atmospheric waters (rain, clouds, aerosol water) are extremely variable from one sample to another (Bianco et al., 2016b; Mopper and Zika, 1987; Triesch et al., 2020; Xu et al., 2019; Yan et al., 2015). It does neither seem feasible nor necessary to perform experiments that consider all possible concentration ranges and ratios. Our brief review below shows that in general AA concentrations are present in micromolar concentrations in cloud water; their distribution likely depends on sources, processing, dilution etc. It should be also noted that not all cloud condensation nuclei contain amino acids, while cloud water concentrations are based on the analysis of bulk water samples. Thus, individual cloud droplets might be much more highly concentrated in amino acids than the bulk cloud water. However, since there are no analytical techniques to date that can routinely determine the solute concentrations in single cloud droplets, we can only take average cloud water concentrations as guidance from our experiments.

Given the multitude of AA sources and distributions and variety in cloud properties, an exactly uniform concentration distribution may not be encountered in any cloud water sample. Our assumption of a uniform distribution could possibly slightly impact the rates of biodegradation, but they should be on the same order of magnitude as we express the rates of biodegradation in mol cell$^{-1}$ h$^{-1}$.

In rain, the total amino acid concentrations vary from 1.1 to 15.5 µM (Mopper and Zika, 1987), from 0.023 to 4.250 µM (Yan et al., 2015), from 1.1 to 10.1 µM (Xu et al., 2019), while in cloud water, it is between 2.7 to 3.1 µM (Bianco et al., 2016b). Considering that between 13 to 18 AA were measured in general, our total AA concentration in this experiment would be around 19 µM as we have included 19 AAs in the solution. This concentration is consistent with what was reported in rain samples, and about five times higher, i.e. less than an order of magnitude, than the concentrations measured in cloud water.

To take this factor of five into account we used an artificial cloud water whose composition was multiplied by 5 compared to what is observed in bulk cloud water samples (Vaïtilingom et al., 2011) and we also used a five-fold concentration for bacteria (Vaïtilingom et al., 2012). So we have respected the concentration ratio of chemical compounds [(main organic and inorganic ions + AA) / number of cells] present in cloud water. In the past we have shown that if the ratio is constant, the rate of biodegradation remains constant in the experiments (Vaïtilingom et al., 2010).

We will modify the Materials and Methods section as follows:

**2.1 Experiments in microcosms**

*The experiments of biotic and abiotic transformation of amino acids were performed in microcosms mimicking cloud conditions at the puy de Dôme station (1465 m). Solar light was fitted to that measured directly under cloudy conditions and the temperature (17°C) was representative of the average temperature in the summer. Rhodococcus enclensis PDD-23b-28, Pseudomonas graminis PDD-13b-3, Pseudomonas syringae PDD-32b-74 and Sphingomonas sp.PDD-32b-11 bacterial strains were chosen because they belong to the most abundant and active bacterial genera in cloud water (Amato et al., 2017; Vaïtilingom et al., 2012). In addition, the complete genome sequences of Rhodococcus enclensis PDD-23b-28, Pseudomonas graminis PDD-13b-3, Pseudomonas syringae PDD-32b-74 have been published recently giving access to their metabolic pathways in more detail (Besaury et al., 2017a, 2017b; Lallement et al., 2017).*  *In this work the total AA concentration used for the incubations was 19 µM as we have included 19 AAs at a concentration of 1µM each in the solution. This concentration is about five times higher than the concentrations measured in cloud water collected as the puy de Dôme station by Bianco et al. (2016a)(the total AA concentration varied from 2.7 to 3.1 µM). To take this factor of five into account we used an artificial cloud water whose composition in inorganic ions, carboxylic acids and amino acids was multiplied by 5 compared to what is observed in clouds ((Vaïtilingom et al., 2011)). We also used a 5X concentration for bacteria (~5×10$^5$ cells mL$^{-1}$) (Vaïtilingom et al., 2012). So we have respected the concentration ratio of chemical compounds [(main organic and inorganic ions*

*+ AA) / number of cells] present in cloud water. In the past we have shown that is the ratio is constant, the rate of biodegradation is constant (Vaïtilingom et al., 2010).*

*All experiments were performed in triplicates.*

Referee Comment:

Concerning the analytical method; the authors used ESI. It is known that ESI is prone the matrix effects (ion suppression) especially in ambient samples containing salt. Therefore, a sample preparation method is often applied, to eliminate disturbing matrix compounds. Did the authors test such effects, as ion suppression for the individual amino acids, for example by comparison of the external calibration to standard addition?

Authors' Response:

We agree that matrix effect can occur using ESI on environmental samples, in which the salt composition and concentration can be very variable. However, in our microcosms experiments, we have used an artificial cloud water medium with a very well-defined composition (Table S1) and we have used exactly the same artificial medium for our external calibration. It is clear from Figure S1 that the signal intensity depends linearly on the AA concentration. The calibrations are performed during the same runs as the experiment analyses. As the salt concentration was identical in the various samples, the matrix effect is the same in all samples. We checked that there is no bias as we can measure the concentrations of the AA at time zero and compare it with the added concentration as we know it (1 µM).

We shall modify this sentence in the Materials and Methods section 2.2.2:

*In order to quantify the amino acid concentrations, calibration curves were established for each experimental series of LC-HRMS analyses using the same artificial cloud medium than in the incubations.*

Referee Comment:

The LOQs seem quite high. How do they compare to other analytical methods used for amino acid analytics? It seems that the LOQ are close to the applied concentration of 1µmol, so did this cause problems in the analytical accuracy? How was the precision (e.g. standard deviation) of the analytical method? As the authors introduce the an-alytical method as a new approach and an improved technique, some further method validation would be necessary in my opinion. How about contaminations? Did you measure blanks and if so, were they considered?

Authors' Response:

In our opinion LOQs are not too high, because we measure concentrations at the beginning of the kinetic experiments of the transformation (initial rates of transformation) and during that period the measured concentrations are above the LOQ. In addition, our experiments are not designed to measure "absolute concentrations", but we measure slopes of $\ln(C_t/C_0) = f(t)$ as demonstrated in Figure S2. As seen in figure S2, the relationship of $\ln(C_t/C_0)$ vs time is very well described by a linear approximation. If the measurements were not sufficiently accurate, the data points would be much

more dispersed. If we compare with the literature in the field of atmospheric sciences, our LOQs are within the same order of magnitude to those described using LC-MS (see table below):

| Amino acid | LOQ[a](nmol L$^{-1}$) | LOQ[b] (µg L$^{-1}$) |
|---|---|---|
| ALA | 20 | 0.2 |
| ARG | 30 | ND |
| ASN | 8 | ND |
| ASP | 20 | 0.2 |
| GLN | 5 | 1.0 |
| GLU | 8 | 0.2 |
| GLY | 40 | 0.2 |
| HIS | 160 | ND |
| ILE+LEU | 10 | 1.0/1.0 |
| LYS | 130 | ND |
| MET | 8 | 1.0 |
| PHE | 4 | 1.0 |
| PRO | 5 | 0.2 |
| SER | 70 | 0.2 |
| THR | 13 | 1.0 |
| TRP | 8 | 1.0 |
| TYR | 7 | ND |
| VAL | 7 | 1.0 |
| CYS | 20 | ND |

a) LOQ determined by LC-MS (direct injection) after extraction of aerosol samples (Helin et al., 2017),b) LOQ determined by UPLC-HRMS (derivatization and concentration by 44 fold) of cloud samples (Triesch et al., 2020). ND: Not determined.

We will add the following text and Table S3 into Section 2.2.2:

*The obtained values of LOD and LOQ were considered to be fit-for-purpose (Table S2) and are consistent with data from the literature ((Helin et al., 2017).*

*We also have calculated the Relative Standard Deviation (RSD = Standard deviation/mean) for each AA based on calibration curves (3 technical replicates). As you can see in the Table S3 these RSD are rather low, ranging from around 0.5% to 10%, except for Valine and Glycine where it can reach 20%. It can be noticed that these RSD due to the LC-MS method are much lower than those due to the transformation experiments, especially for biotransformation where there are biological variations (see error bars in Figure 1 and 2)*

Table S3: Relative standard deviation (RSD = Standard deviation/mean) for each AA based on calibration curves (3 technical replicates).

| | Relative Standard Deviation (RSD = Standard deviation/mean) | | |
|---|---|---|---|
| Amino acid | 0.1 µM (n = 3) | 0.5 µM (n = 3) | 1 µM (n = 3) |
| ALA | | 0.71% | 3.61% |
| ARG | 0.83% | 1.96% | 1.56% |
| ASN | 5.23% | 4.92% | 3.63% |
| ASP | | 10.77% | 5.96% |

| | | | |
|---|---|---|---|
| GLN | 4.19% | 4.37% | 3.20% |
| GLU | 3.77% | 2.89% | 3.92% |
| GLY | | | 21.39% |
| HIS | 0.62% | 0.89% | 1.22% |
| ILE | 4.48% | 0.48% | 0.59% |
| LYS | 6.64% | 1.96% | 1.50% |
| MET | 4.49% | 4.35% | 6.38% |
| PHE | 4.63% | 1.68% | 1.02% |
| PRO | 11.67% | 5.08% | 1.28% |
| SER | 14.34% | 3.06% | 3.20% |
| THR | 14.15% | 3.67% | 1.06% |
| TRP | 7.00% | 1.67% | 1.75% |
| TYR | 0.94% | 1.81% | 1.15% |
| VAL | 17.94% | 2.98% | 11.41% |

Of course, blanks were made for each series of runs. They consisted of using the artificial cloud medium without AAs. No signals corresponding to AA are detected under these conditions.

As we have introduced this new Table S3, the previous Tables S3 and S4 will be renamed Tables S4 and S5

**Table S4**: Rate constants for 18 amino acids for the OH, $O_3$ and $^1O_2$ reactions

**Table S5:** Selected experimental studies of amino acid oxidation by various oxidants. Note that the experimental conditions were not necessarily atmospherically-relevant. Products are only listed to demonstrate the wide variety of possible reaction pathways and products.

Referee Comment:

Finally; would your analytical method (without pre-concentration and sample preparation) be applicable for measuring amino acids in ambient marine samples?

Authors' Response:

Using this method for marine samples may cause some problems due to the much higher salt concentrations (~0.1 M) and the lower AA concentration than encountered in cloud water where these concentrations are in the range of milli- to micromolar, respectively. In addition, to prevent matrix effects, we recommend to use the addition of a standard method for calibration and not an external calibration.

Referee Comment:

Chapter 2.1.: The authors explained that the strains were chosen because they are the most abundant and active bacteria in cloud water. Are there more information on these strains available, that might be used to explain their different behaviour towards the individual amino acids?

Authors' Response:

We have used these strains in many previous studies to explore their biodegradation of a variety of organic compounds, e.g. small carboxylic acids (Vaïtilingom et al., 2010,2011) or phenol (Jaber et al., 2020). We could not observe such high difference between the biological activities of these strains towards these organics. Thus, the different behavior of the different strains towards AAs cannot be explained despite the fact that they belong to different genera (Figure S5).

Referee Comment:

Chapter 3.1.1: Interestingly, the efficiencies of the different strains are very variable among each other and concerning the different amino acids. The authors mentioned that all amino acids were mixed together in the experiment. I was wondering if you also performed these experiments with single amino acids? This might be interesting especially regarding the net production of GLY that is certainly a product from the degradation of other amino acids.

Authors' Response:

We chose to perform the experiments with this mixture of AA in a medium mimicking the cloud medium to be as close of possible to realistic atmospheric conditions. Working with single AA could be interesting but very time consuming and would not reflect real cloud conditions.

Referee Comment:

Chapter 3.1.2.: The manuscript often refers to the Figures S3 and S4 which seem to be crucial for following and understanding the text. As the manuscript does not contain many Figures, maybe transfer them to the main part? An alternative could be to highlight the amino acids that have the same metabolic pathway in Figure 1(instead of Figure S4). The statement that the "blue box" amino acids exhibit the same behaviour regarding their biodegradation is difficult to see in Figure S4 and a "zoom in" would be required. Actually, it seems that GLY shows quite a different behavior, not in line with the other "blue box" amino acids. Also the "green box" amino acids are difficult to see (Fig. S4). For the "purple box" amino acids; the mentioned strong similarities are not obvious from Fig. S4. The 23b-28 strain seems to be much stronger for ASN compared to ALA. Please re-think the way of showing the similarities and maybe find a clearer way to present similarities and differences for the metabolomic-groups amino acids and their response to the different strains.

Authors' Response:

We agree with the referee that it is not easy to look at the different figures in the main text and in the SI. Actually we really thought at the various possibilities and decided that the one we chose was the clearest. We prefer to keep the main manuscript concise and only show the essential results and only provide additional details in the SI.

Referee Comment:

Chapter 3.1.3 Are there any more detailed explanation theories why these different strains exhibit such different behaviours? To what properties could that be related? On page 9, line 24 the authors mention that the AA biodegradation could be linked to the phylogeny of the bacterial strains. Could you give some more explanation (to non-biologists) about this?

Authors' Response:

As explained in the text, metabolic pathways are rather similar for all the living organisms, however the metabolic fluxes (i.e. rates of transformation of metabolites by each enzyme) can be modulated by the environmental conditions and the type of organisms (namely their phylogeny). In our experiments, the environmental conditions are the same for all the four studied strains, so the observed differences are only due to their phylogeny. We can see in Figure S3 that the two *Pseudomonas* strains (closely related from a phylogenetic point of view as they belong to the same genus "*Pseudomonas*", and same class "$\gamma$-*Proteobacteria* ") have a closer behavior that the other strains. However we are not able to understand what is the direct connection between the phylogeny and the biological activity towards AA, and thus we are not able to predict the activity of a strain looking at its phylogeny.

 To make clearer the notion of phylogeny we propose to add this text in the SI under Figure S5

*An example of phylogenetic classification is given bellow*

*Phylum--$\rightarrow$Class$\rightarrow$Genus$\rightarrow$species$\rightarrow$strain number*

*Proteobacteria$\rightarrow$$\gamma$-Proteobacteria$\rightarrow$Pseudomonas$\rightarrow$graminis$\rightarrow$ PDD-13b-3*

Referee Comment:

Chapter 3.3.1: I wonder how relevant singlet oxygen is for diluted systems. (lifetime?) Is the sink for singlet oxygen considered in the rates (Fig 3)?

Authors' Response:

There are several studies that have reported a steady state singlet oxygen concentration in fog and cloud waters on the order of $10^{-14} - 10^{-12}$ M (Faust and Allen, 1992; Kaur and Anastasio, 2017), similar to concentrations found in surface water (Faust and Allen, 1992). This is about two orders of magnitude higher than steady-state concentrations OH radical in the atmospheric aqueous phases. OH is considered the main oxidant in the atmospheric multiphase (gas + aqueous) system because of its high reactivity towards many organic and inorganic compounds. The lifetime of singlet oxygen is longer than that of the OH radical in water as it is more selective towards reactants (Kaur and Anastasio, 2017).

Given the high rates of production and loss processes of the radical species (OH and $^1O_2$) that result in stable steady-state concentrations, we only considered these concentrations to estimate the loss rates of the amino acids. This approach implies (pseudo) first order kinetics as it has been used in

many previous studies that estimated the chemical lifetime of various compounds in the atmosphere (and other media), e.g (McGregor and Anastasio, 2001; Triesch et al., 2020), or more general in standard atmospheric chemistry books (Seinfeld and Pandis, 2006). As explained in the text already, we refrained from presenting our results in terms of lifetimes as production rates would result in negative values which are clearly meaningless.

Figure 2 shows that degradation and formation happens for the individual amino acids. As a general question and also related to Fig. 2: Can any mechanisms for the formation/degradation of the individual amino acids be derived from that?

Authors' Response:

We cannot give any additional reliable information on the mechanism of the amino acid decay and formation as currently there is no sufficient mechanistic information available. The studies summarized in Table S4 give some hints that the oxidation of amino acids can possibly lead to the formation of other amino acids. However, since these studies were neither performed under conditions similar to those in our experiments (and thus to those as relevant for cloud water), nor were any yields or branching ratios reported, any conclusions on the transformation of AAs would be speculative. We hope that our study motivates laboratory experiments in the future that investigate in detail the mechanisms, yields, branching ratios and time scales of such conversions so that ultimately a figure as the 'chemical equivalent' to Figure S3 could be created, i.e. with the chemical instead of metabolic routes included.

Chapter 3.3.2 and Figure 4: This chapter deals with the comparison of the biotic and the abiotic pathway. They are shown in Fig.3. While some exemplary comparisons areC3made between both pathways (Page 11, Line 30 - Page 12 Line 6) I miss some real conclusions here. In addition, Figure 3 is difficult to understand and not well discussed; some more details might be helpful to understand the outcome of Fig. 3.

Authors' Response:

We assume that the referee's comment refers only to Figure 3 here as there is no Figure 4 in the original manuscript. We agree with the referee that the description and discussion of Figure 3 was rather short. We will modify Section 3.3.2 as follows:

- We add an index 'bio' to the left-hand term in Equation 2 so it reads

$$\left(\frac{d[AA]}{dt}\right)_{bio} = -0.063\,R_{23b28} \cdot 1.91 - 0.162\,R_{32b11} \cdot 1.91 - \frac{29.8}{2}R_{23b28b} \cdot 1.91 - \frac{29.8}{2}R_{13b2} \cdot 1.91$$

- We modify the last paragraph in Section 3.3.2 (new text in green):

*The three rates, i.e. the biodegradation (Eq.-2) and photochemical (Eq.-3) rates as derived from the experiments, and the kinetic loss rates based on chemical kinetics (Eq.-4), respectively, are compared in Figure 3 for teach of the 18 amino acids. For some of the acids (ALA, GLU, THR) the predicted losses by OH from both approaches (photochemical experiments (red dashed bars) and based on OH kinetic data (solid dark red bars)) are similar. Thus, we can conclude that these acids are oxidized to products other than amino acids and that the approximation of their loss rates by Equation 4 is justified, as it has been done in previous studies, e.g. (McGregor and Anastasio, 2001; Triesch et al., 2020). For*

*several other amino acids (e.g. ARG, GLN, LYS, SER, and THR) there is a large discrepancy in the observed trends of the predicted chemical loss rates and the ones observed in the photochemical experiments. The latter ones have positive values, i.e. they indicate a net production rather than a net loss. While we cannot conclude on the exact conversion and formation mechanisms of these acids based on our experiments, it is evident that the assumption of a net loss underestimates the lifetime of these acids as they do not only have chemical sinks but also sources in the atmospheric aqueous phase. As also reflected in Figure 1, such net production is only seen for ASP and GLY for biotic processes.*

* The comparison of the predicted role of the three oxidants in cloud water (OH, $O_3$, $^1O_2$) reveals for some AAs, the oxidation by ozone might contribute significantly more to their loss than the other two oxidants (light red bars; note the logarithmic scale, i.e. the contributions of the ozone reactions to the total predicted loss exceeds those by other oxidants by far).*

*For several of the acids (e.g. ALA, ASN, GLU, PRO, VAL), biotransformation is predicted to exceed the loss by chemical reactions , for the bacteria cell and oxidant concentrations considered here. Given that the ratios of bacteria cells/radicals in our estimate here are similar to those as encountered in cloud water, it may be concluded that both types of pathways might compete in the atmosphere. Similar conclusions were qualitatively drawn based on ambient measurement in a recent study (Zhu et al., 2020). However, the exact contributions of biotic and abiotic pathways to the loss and conversion of amino acids will depend on the cell concentrations of the different bacteria strains, their distribution among cloud droplets, and oxidant levels.*

**

Chapter 4: The conclusions are well written. The authors summarize that the so far only degradation (losses) of amino acids but not production (transformation into each other) was considered. However, I was struggling with the following sentence: "Our study qualitatively suggests that the sources and distribution of amino acids in the atmospheric particle and aqueous phases can be modified by metabolic and chemical transformation pathways." -> Could the authors derive more precise conclusions here? I understood it was the aim to show HOW the two pathways (biotic, abiotic) contribute. I was wondering if the authors could finally comment on the relative importance of the biotic and the abiotic pathway e.g. which seems to be the more important way?

Authors' Response:

We thank the referee for this comment. Our study is the first one to suggest based on lab studies the formation and conversion of amino acids by not only biotic but also by chemical processes. Overall, we can conclude that both types of processes might be similarly important for many of the amino acids as shown in Figure 3 under atmospheric conditions. The exact rates will depend on the distribution of the radicals and bacteria cells throughout the cloud droplet population. Based on the analysis of cloud water samples (Vaïtilingom et al., 2013) and recent model studies (Khaled et al., 2020), it can be hypothesized that the low fraction of cloud droplets that contain bacteria cells might translate into very non-linear overall loss rates of non-volatile compounds (such as amino acids).

However, given the large variability in the atmosphere of cloud properties, bacteria diversity and cell concentrations, oxidant concentrations (e.g. depending on air mass characteristics, photochemical activity etc) and amino acid sources and distributions (cf e.g. references cited in the introduction of our manuscript), we cannot perform a global estimate of the relative importance of biotic versus abiotic amino acid processes.

Minor referee comments:

- There are several typos e.g. page 2 line 11 (C.L-1), sometimes the chemicals / amino acids are written with capital letter, sometimes with small letters (e.g.Table S4).

- Empty spaces are missing and the formulas in eq. 2-4 are not represented right.

Authors' Response: We will fix the formatting of the equations and will make sure that are correct in the uploaded pdf files.

- In addition, the reference style needs revisions (e.g. page 16, line 44-45, page 17, line11, page 19, line 25.

Authors' Response: We will make sure to use the Copernicus template for reference formatting so that the references in the text are correct.

- Table S3: There are missing references (for GLU, GLY, SER...).

Authors' Response: We will add the missing references.

- At what temperature was the rate constant obtained?

Authors' Response: Whenever possible we chose rate constants at or near room temperature. We will add this information to the table caption.

- Concerning the Data availability I'd strongly recommend to upload the data in a public database such as PANGAEA or similar.

Authors' Response: Data are available upon request

- Author contributions: I was surprised that "SJ", as the first author, did not "write the manuscript"?

Authors' Response: In our team, the first author is the one who made the largest contribution to the work, here it is considered for the experimental work which is very demanding. She also read and corrected the manuscript (as noticed in the text).

**References**

Amato, P., Joly, M., Besaury, L., Oudart, A., Taib, N., Moné, A. I., Deguillaume, L., Delort, A. M. and Debroas, D.: Active microorganisms thrive among extremely diverse communities in cloud water, PLoS One, doi:10.1371/journal.pone.0182869, 2017.

Besaury, L., Amato, P., Wirgot, N., Sancelme, M. and Delort, A. M.: Draft genome sequence of Pseudomonas graminis PDD-13b-3, a model strain isolated from cloud water, Genome Announc., doi:10.1128/genomeA.00464-17, 2017a.

Besaury, L., Amato, P., Sancelme, M. and Delort, A. M.: Draft genome sequence of Pseudomonas syringae PDD-32b-74, a model strain for ice-nucleation studies in the atmosphere, Genome Announc., doi:10.1128/genomeA.00742-17, 2017b.

Bianco, A., Voyard, G., Deguillaume, L., Mailhot, G. and Brigante, M.: Improving the characterization of dissolved organic carbon in cloud water: Amino acids and their impact on the oxidant capacity, , 6, 37420, doi:10.1038/srep37420 https://www.nature.com/articles/srep37420#supplementary-information, 2016a.

Bianco, A., Passananti, M., Deguillaume, L., Mailhot, G. and Brigante, M.: Tryptophan and tryptophan-like substances in cloud water: Occurrence and photochemical fate, Atmos. Environ., 137, 53–61, doi:10.1016/j.atmosenv.2016.04.034, 2016b.

Deguillaume, L., Charbouillot, T., Joly, M., Vaïtilingom, M., Parazols, M., Marinoni, A., Amato, P., Delort, A. M., Vinatier, V., Flossmann, A., Chaumerliac, N., Pichon, J. M., Houdier, S., Laj, P., Sellegri, K., Colomb, A., Brigante, M. and Mailhot, G.: Classification of clouds sampled at the puy de Dôme (France) based on 10 yr of monitoring of their physicochemical properties, Atmos. Chem. Phys., 14(3), 1485–1506, doi:10.5194/acp-14-1485-2014, 2014.

Faust, B. C. and Allen, J. M.: Aqueous-phase photochemical sources of peroxyl radicals and singlet molecular oxygen in clouds and fog, J. Geophys. Res. Atmos., 97(D12), 12913–12926, doi:10.1029/92JD00843, 1992.

Helin, A., Sietiö, O.-M., Heinonsalo, J., Bäck, J., Riekkola, M.-L. and Parshintsev, J.: Characterization of free amino acids, bacteria and fungi in size-segregated atmospheric aerosols in boreal forest: seasonal patterns, abundances and size distributions, Atmos. Chem. Phys., 17(21), 13089–13101, doi:10.5194/acp-17-13089-2017, 2017.

Jaber, S., Joly, M., Brissy, M., Leremboure, M., Khaled, A., Ervens, B. and Delort, A.-M.: Biotic and abiotic transformation of amino acids in cloud water: Experimental studies and atmospheric implications, Biogeosciences Discuss., 2020, 1–27, doi:10.5194/bg-2020-250, 2020.

Kaur, R. and Anastasio, C.: Light absorption and the photoformation of hydroxyl radical and singlet oxygen in fog waters, Atmos. Environ., 164, 387–397, doi:https://doi.org/10.1016/j.atmosenv.2017.06.006, 2017.

Khaled, A., Zhang, M., Amato, P., Delort, A.-M. and Ervens, B.: Biodegradation by bacteria in clouds: An underestimated sink for some organics in the atmospheric multiphase system, Atmos. Chem. Phys. Discuss., 2020, 1–32, doi:10.5194/acp-2020-778, 2020.

Lallement, A., Besaury, L., Eyheraguibel, B., Amato, P., Sancelme, M., Mailhot, G. and Delort, A. M.: Draft Genome Sequence of Rhodococcus enclensis 23b-28, a Model Strain Isolated from Cloud Water, Genome Announc., 5(43), e01199-17, doi:10.1128/genomeA.01199-17, 2017.

McGregor, K. G. and Anastasio, C.: Chemistry of fog waters in California's Central Valley: 2. Photochemical transformations of amino acids and alkyl amines, Atmos. Environ., 35(6), 1091–1104, doi:Doi: 10.1016/s1352-2310(00)00282-x, 2001.

Mopper, K. and Zika, R. G.: Free amino acids in marine rains: evidence for oxidation and potential role in nitrogen cycling, Nature, 325(6101), 246–249, doi:10.1038/325246a0, 1987.

Seinfeld, J. H. and Pandis, S. N.: Atmospheric Chemistry and Physics - From air pollution to climate change, 2nd ed., edited by I. John Wiley and Sons, John Wiley & Sons, Inc., Hoboken, New Jersey., 2006.

Triesch, N., van Pinxteren, M., Engel, A. and Herrmann, H.: Concerted measurements of free amino acids at the Cape Verde Islands: High enrichments in submicron sea spray aerosol particles and cloud droplets, Atmos. Chem. Phys. Discuss., 2020, 1–24, doi:10.5194/acp-2019-976, 2020.

Vaïtilingom, M., Amato, P., Sancelme, M., Laj, P., Leriche, M. and Delort, A.-M.: Contribution of Microbial Activity to Carbon Chemistry in Clouds, Appl. Environ. Microbiol., 76(1), 23–29, doi:10.1128/AEM.01127-09, 2010.

Vaïtilingom, M., Charbouillot, T., Deguillaume, L., Maisonobe, R., Parazols, M., Amato, P., Sancelme, M. and Delort, A. M.: Atmospheric chemistry of carboxylic acids: microbial implication versus photochemistry, Atmos. Chem. Phys., 11(16), 8721–8733, doi:10.5194/acp-11-8721-2011, 2011.

Vaïtilingom, M., Attard, E., Gaiani, N., Sancelme, M., Deguillaume, L., Flossmann, A. I., Amato, P. and Delort, A.-M.: Long-term features of cloud microbiology at the puy de Dôme (France), Atmos. Environ., 56(0), 88–100, doi:http://dx.doi.org/10.1016/j.atmosenv.2012.03.072, 2012.

Vaïtilingom, M., Deguillaume, L., Vinatier, V., Sancelme, M., Amato, P., Chaumerliac, N. and Delort, A.-M.: Potential impact of microbial activity on the oxidant capacity and organic carbon budget in clouds, Proc. Natl. Acad. Sci., 110(2), 559–564, doi:10.1073/pnas.1205743110, 2013.

Xu, Y., Wu, D., Xiao, H. and Zhou, J.: Dissolved hydrolyzed amino acids in precipitation in suburban Guiyang, southwestern China: Seasonal variations and potential atmospheric processes, Atmos. Environ., 211, 247–255, doi:https://doi.org/10.1016/j.atmosenv.2019.05.011, 2019.

Yan, G., Kim, G., Kim, J., Jeong, Y.-S. and Kim, Y. Il: Dissolved total hydrolyzable enantiomeric amino acids in precipitation: Implications on bacterial contributions to atmospheric organic matter, Geochim. Cosmochim. Acta, 153, 1–14, doi:10.1016/j.gca.2015.01.005, 2015.

Zhu, R., Xiao, H.-Y., Luo, L., Xiao, H., Wen, Z., Zhu, Y., Fang, X., Pan, Y. and Chen, Z.: Measurement report: Amino acids in fine and coarse atmospheric aerosol: concentrations, compositions, sources and possible bacterial degradation state, Atmos. Chem. Phys. Discuss., 2020, 1–30, doi:10.5194/acp-2020-534, 2020.

---

## Author Comment (AC2) · 1 Oct 2020

Referee #2

Referee Comment:

In this work the authors conducted microcosm experiments with the aim of differentiating the roles of biotic and abiotic transformation of free amino acids in cloud water. In the experiment, they utilized 19 types of amino acids, four bacterial strains or photo-bioreactors, attempting to mimic ambient cloud conditions. With the kinetic loss/production data of amino acids, they concluded that previous studies may have overestimated the abiotic degradation rates of amino acids, and future modeling efforts should take the biotic and abiotic transformation of amino acids into account. Overall, I think the authors did a solid job in terms of writing and offered an interesting dataset in amino acid dynamics in the field of cloud chemistry. The data seem to be sound. However, I have some concerns the authors need to address before publication.

Authors' Response:

We thank the referee for their positive evaluation of our manuscript. We address all comments point-by-point below.

Referee Comment:

The authors claimed that they developed a new analytical technique that can analyze amino acids, without the need of any preparation such as derivatization, using UPLCHRMS. This is nice effort, but I am surprised that the incubation medium they used, containing mM of ions such as Na, Ca an K etc. (Table S1), which would directly impact ion source and ionization process, was directly injected to the LC-MS. I don't think any mass spectrometry expert would be happy if you inject such a solution to the instrument.

Authors' Response:

As you can see in Table 1 the concentrations are not in mM but in µM, except for $Na^+$ which is 1mM, therefore we did not find any problem with LC-MS. We are far from concentrations encountered in ocean water samples for instance where concentrations are in the range of 0.1M. As shown in Figure S1 linear plots are obtained for the calibration curves. Also we have calculated the Relative standard deviation (RSD = Standard deviation/mean) for each AA based on calibration curves (3 technical replicates). As you can see in the Table S3 these RSD are rather low, ranging from around 0.5% to 10%, except for Valine and Glycine where it can reach 20%. Finally, we know the initial concentration of the AA (1µM) and we do find this concentration in our measurements. Our conclusion is this that this method is suited for measuring AA concentration in this medium, in addition the obtained values for LOD and LOQ are within the same range of order than those reported in the literature.

We will add the following text and Table S3 into Section 2.2.2:

The obtained values of LOD and LOQ were considered to be fit-for-purpose (Table S2) *and are consistent with data from the literature (*(Helin et al., 2017)*.*

*We also have calculated the Relative standard deviation (RSD = Standard deviation/mean) for each AA based on calibration curves (3 technical replicates). As you can see in the Table S3 these RSD are rather low, ranging from around 0.5% to 10%, except for Valine and Glycine where it can reach 20%. It can be noticed that these RSD due to the LC-MS method are much lower than those due to the transformation experiments, especially for biotransformation where there are biological variations (see error bars in Figure 1 and 2).*

Table S3: Relative standard deviation (RSD = Standard deviation/mean) for each AA based on calibration curves (3 technical replicates)

| | Relative standard deviation (RSD = Standard deviation/mean) | | |
|---|---|---|---|
| Amino acid | 0.1 µM (n = 3) | 0.5 µM (n = 3) | 1 µM (n = 3) |
| ALA | | 0.71% | 3.61% |
| ARG | 0.83% | 1.96% | 1.56% |
| ASN | 5.23% | 4.92% | 3.63% |
| ASP | | 10.77% | 5.96% |
| GLN | 4.19% | 4.37% | 3.20% |
| GLU | 3.77% | 2.89% | 3.92% |
| GLY | | | 21.39% |
| HIS | 0.62% | 0.89% | 1.22% |
| ILE | 4.48% | 0.48% | 0.59% |
| LYS | 6.64% | 1.96% | 1.50% |
| MET | 4.49% | 4.35% | 6.38% |
| PHE | 4.63% | 1.68% | 1.02% |
| PRO | 11.67% | 5.08% | 1.28% |
| SER | 14.34% | 3.06% | 3.20% |
| THR | 14.15% | 3.67% | 1.06% |
| TRP | 7.00% | 1.67% | 1.75% |
| TYR | 0.94% | 1.81% | 1.15% |
| VAL | 17.94% | 2.98% | 11.41% |

As we have introduced this new Table S3, the previous Tables S3 and S4 will be renamed Tables S4 and S5

***Table S4****: Rate constants for 18 amino acids for the OH, O$_3$ and $^1$O$_2$ reactions*

***Table S5:*** *Selected experimental studies of amino acid oxidation by various oxidants. Note that the experimental conditions were not necessarily atmospherically-relevant. Products are only listed to demonstrate the wide variety of possible reaction pathways and products.*

Referee Comment:

I am wondering what's the inject volume they used and how they can maintain a consistent sensitivity with such high ion strength solution (or how long).

Authors' Response: The injection volume was 5µL which is very low and does not induce any problem.

We shall add this information in the Material and Methods section 2.2.1

*The volume of injection was 5µL.*

Referee Comment:

In addition, external calibration curves were used to quantify the amino acids in cloud water medium. Did you use the same water medium for the standards? If not, this might be a problem with the matrix effect.

Authors' Response: We used the same medium for the standards.

We shall add this information in the Material and Methods section 2.2.2

In order to quantify the amino acid concentrations, calibration curves were established for each experimental series of LC-HRMS analyses *using the same artificial cloud medium than in the incubations.*

Referee Comment:

It may be also helpful to show the LC-MS chromatograms in the supplementary section. The bottom line is that more info is needed for this new approach you developed.

Authors' Response:

As explained in the 2.2.1 section, the ions were selected using the SIM (Selected Ion Monitoring) mode for each AA so that the raw LC-MS chromatograms are not of great interest (not used for quantification). In addition, as Q-Orbitrap™ was used, the extracted masses are very precise.

Referee Comment:

I like the experimental approach, such as clearly separating abiotic and biotic factors, and using free amino acids and single bacterial strains, which allowed you to tease out the convoluted factors observed in field samples. However, the authors need to realize/justify their experimental conditions which I think are far away from those of the field, thus more discussion is needed. For example, they used 1 uM of 19 types of amino acids, which represent 19uM amino acids or 684 ug C/L (assuming 3 C per amino acid); in contrast, the cloud water only contained 2.4-74.3 ug C/L, cited from the Introduction of the manuscript.

Authors' Response:

Cloud and fog water usually contains several milligrams of carbon per liter, a small fraction of which is composed of amino acids. As cited in the introduction, concentrations of up to 757 µg C L$^{-1}$ amino acids have been identified in cloud water; thus, we do not think that our assumptions are unrealistic. It should be also noted that not all cloud condensation nuclei contain amino acids, while cloud water concentrations are based on the analysis of bulk water samples. Thus, individual cloud droplets might be much more highly concentrated in amino acids than the bulk cloud water. However, since there are no analytical techniques to date that can routinely determine the solute concentrations in single cloud droplets, we can only take average cloud water concentrations as guidance from our experiments.

In addition, if we express the AA concentrations in molarity, the total amino acid in rain varied from 1.1 to 15.5 µM (Mopper and Zika, 1987), from 0.023 to 4.250 µM (Yan et al., 2015), from 1.1 to 10.1 µM (Xu et al., 2019), while in cloud it was from 2.7 to 3.1 µM (Bianco et al., 2016b). Considering that between 13 to 18 AA were measured in general, our total AA concentration in this experiment would be around 19 µM as we have included 19 AAs in the solution. This concentration is consistent with what was reported in rain samples, and about five times higher than the concentrations measured in clouds.

To take this factor of five into account we used an artificial cloud water whose composition was multiplied by 5 compared to what is observed in clouds (Vaïtilingom et al., 2011) and we also used a five-fold concentration for bacteria (Vaïtilingom et al., 2012). So we have respected the concentration ratio of chemical compounds [(main organic and inorganic ions + AA) / number of cells] present in cloud water. In the past we have shown that is the ratio is constant, the rate of biodegradation is constant (Vaïtilingom et al., 2010).

We shall modify the text in section 2.1 to explain and justify better our experimental conditions.

**2.1 Experiments in microcosms**

*The experiments of biotic and abiotic transformation of amino acids were performed in microcosms mimicking cloud conditions at the puy de Dôme station (1465 m). Solar light was fitted to that measured directly under cloudy conditions and the temperature (17°C) was representative of the average temperature in the summer. Rhodococcus enclensis PDD-23b-28, Pseudomonas graminis PDD-13b-3, Pseudomonas syringae PDD-32b-74 and Sphingomonas sp.PDD-32b-11 bacterial strains were chosen because they belong to the most abundant and active bacterial genera in cloud water (Amato et al., 2017; Vaïtilingom et al., 2012). In addition, the complete genome sequences of Rhodococcus enclensis PDD-23b-28, Pseudomonas graminis PDD-13b-3, Pseudomonas syringae PDD-32b-74 have been published recently giving access to their metabolic pathways in more detail (Besaury et al., 2017a, 2017b; Lallement et al., 2017).  In this work the total AA concentration used for the incubations was 19 µM as we have*

*included 19 AAs at a concentration of 1µM each in the solution. This concentration is about five times higher than the concentrations measured in cloud water collected as the puy de Dôme station by Bianco et al. (2016a)(the total AA concentration varied from 2.7 to 3.1 µM). To take this factor of five into account we used an artificial cloud water whose composition in inorganic ions, carboxylic acids and amino acids was multiplied by 5 compared to what is observed in clouds (Vaïtilingom et al., 2011). We also used a 5X concentration for bacteria (~5×10⁵ cells mL⁻¹) (Vaïtilingom et al., 2012). So we have respected the concentration ratio of chemical compounds [(main organic and inorganic ions + AA) / number of cells] present in cloud water. In the past we have shown that is the ratio is constant, the rate of biodegradation is constant (Vaïtilingom et al., 2010).*

*All experiments were performed in triplicates.*

Referee Comment:

Your rate calculation is dependent on the concentration, thus the extremely high concentrations you used could have led to a conclusion not relevant to the field (the rate constant could also change depend on how the bacteria take up the substrate).

Authors' Response:

Generally, the rate calculations are only dependent on the bacteria cell and oxidant concentrations, respectively, cf equations 2 – 4. There are several studies that corroborate these cell concentrations in cloud water (Hu et al., 2018; Sattler et al., 2001; Vaïtilingom et al., 2013). Given that bacteria are very efficient cloud condensation nuclei (Zhang et al., 2020), the more abundant measurements of ambient particle concentrations ($\sim 10^3 - 10^5$ cm$^{-3}$) can be used to infer similar cell concentrations in cloud water.

OH concentrations in cloud water have been indirectly determined by measurements (Arakaki et al., 2013; Bianco et al., 2015) and by numerous model studies (Ervens et al., 2003; Herrmann et al., 2010; Tilgner et al., 2013). While there are significantly fewer estimates available for singlet oxygen, the few available studies agree that it is about two orders of magnitude higher than OH (Faust and Allen, 1992)(Kaur and Anastasio, 2017). The ozone concentration can be calculated based on its Henry's law constant as cloud water's ionic strength is sufficiently low (< millimolar) to approximate it as an ideal solution. Thus, we are confident that the chemical rate constants and resulting rates are not influenced by the medium as it is also assumed in all cloud chemistry models.

In addition, when bacteria are just harvested from culture medium, in a way they have been pre-trained to take up labile organic matter rapidly, thus the biotic loss rate you obtained could have been overestimated.

Authors' Response:

Usually bacteria have to adapt their metabolism only to new substances which are typically xenobiotics (ex: Phenol) or to compounds which are not part of the central metabolism (ex:

formaldehyde, formate). On the contrary AA are essential substrates for bacteria and are metabolized in the central metabolism, they do not need to adapt their metabolism to these substrates. We have shown in the past that when we incubate real cloud samples, bacteria can grow in this medium showing they do use these substrates (Amato et al., 2007). In our opinion, the only important point is that these rates will depend on the type of bacteria, so on the biodiversity in cloud water that likely varies from one cloud to the other. These rates could also change according to different atmospheric scenarios. More work is indeed needed to have a clear overview of what happens in real clouds, notably biodegradation and photo-degradation rates should be measured with real cloud samples to evaluate the variability of these degradation rates. This is why we wrote in the conclusion *"Our study highlights the need for further mechanistic investigations of the biotic (metabolic) and abiotic (chemical) transformations of amino acids under conditions relevant for the atmospheric aqueous phases (clouds, fogs, aerosols)."* (section 4., last paragraph).

Referee Comment:

Bacteria in the cloud water, on the contrary, may not be that active often due to substrate limitation.

Authors' Response:

In cloud medium the concentration of AA is rather low but the concentration of bacteria is also low ($10^5$cells.$L^{-1}$), so in our opinion there is no substrate limitation. This assumption is supported by the following elements: i) in these experiments we have respected the ratio cells/ AA concentrations observed in clouds and have been able to measure rates of biotransformation, ii) we have proven that bacteria can use AA as substrates in incubations with real cloud water containing endogenous bacteria and AAs because they can produce proteins and other cellular component allowing their growth in this medium (Amato et al, ACP, 2007), iii) a recent metatranscriptomic study performed directly in cloud water, showed the presence of transcripts of genes coding for AA biodegradation and synthesis (Amato et al, Sci. Rep. 2019). This is a proof of *in situ* activity of cloud bacteria in clouds.

We shall modify the introduction as follows:

*In cloud water, the biodegradation and biosynthesis of AAs is suspected to occur as i) it was shown that bacteria can use AA as substrates in incubations with real cloud water containing endogenous bacteria and AAs because they can produce proteins and other cellular component allowing their growth in this medium (Amato et al., 2007), iii) a recent metatranscriptomic study performed directly in cloud water, showed the presence of transcripts of genes coding for AA biodegradation and synthesis (Amato et al., 2019). This is a proof of in situ activity of bacteria in clouds.* However, no data exist about the biotransformation rates  of AAs in cloud water.

Referee Comment:

Similarly, the abiotic transformation rates would be different when you have an organic matrix present, like cloud water. As mentioned in the Introduction, organic matter in the cloud water is complicated, including many different compounds, which may include quenchers and photosensitizers, the rates you obtained may not represent those of field. I think all these need to be

factored in when you try to argue against previous studies, or apply these data to the field. I would like to see more discussion along these perspectives.

Authors' Response:

We respectfully disagree that cloud water represents an organic medium. The referee is right that it contains many organics with a very complex and variable composition but yet the main solvent is water with dissolved solutes at millimolar or even lower concentrations (e.g., depending on cloud droplet size). This is different in water associated with aerosol particles, i.e. outside of clouds, where indeed ionic strengths of several moles per Liter or more can be present and organic and aqueous phases may be separated due to different solute activities.

The oxidant concentrations used in our estimates are the steady state concentrations. Several studies reported a steady state singlet oxygen concentration in fog and cloud waters on the order of $10^{-14} - 10^{-12}$ M (Faust and Allen, 1992; Kaur and Anastasio, 2017), similar to concentrations found in surface water (Faust and Allen, 1992), and about two orders of magnitude higher than the OH radical which is considered the main oxidant in the atmospheric multiphase (gas + aqueous) system because of its high reactivity towards many organic and inorganic compounds. The lifetime of singlet oxygen is longer than that of the OH radical in water as it is more selective towards reactants (Kaur and Anastasio, 2017).

These steady-state concentrations are a result of the high production and loss rates of these oxidants (OH and $^1O_2$) from multiple pathways. Such pathways may include processes with quenchers or photosensitizers. However, given that these oxidant concentrations were determined in real cloud, fog and surface waters, the use of the resulting steady-state concentrations to estimate loss rates is common and justified.

The referee is right that there might be other loss processes for amino acids or other organics. However, there is no doubt in the atmospheric chemistry community that the OH radical is the most powerful and most important oxidant both in the gas (Seinfeld and Pandis, 2006) and aqueous phases (Ervens et al., 2003). The overall importance of ozone and singlet oxygen is lower as they are both more selective towards reactants. As they do react with amino acids, however, we consider their loss rates to our estimate in Figure 3.

Referee Comment:

Many places in the Introduction, there are too many references which kind of stops the flow. I would suggest you only choose the key ones.

Authors' Response:

It is rather difficult to choose key references as they are all essential to provide the scientific background for the manuscript's topic. Therefore, we prefer to keep all the cited references.

Minor referee comments

Page 3 line 28: "biotranform"? You meant: ": : :shown to biotransform: : :"

Page 7 line 25: no need to list all these amino acids here.

Page 8 line 3: should be "due to"

Page 8 line 7: should be "bacterial strain"

Page 8 line 9: I think "production" is a better word than "synthesis" here.

Page 9 line 26: delete "in the experiments", redundant

Page 11 line 5: delete the ": : :"

Page 11 lines 10&17: the equations did not show up right. It might have something to do with the formatting.

Page 12 line 8: delete "It is obvious that"

Page 12 line 30: right, but as I mentioned before, your experimental conditions may not be that relevant, either.

Authors' Response: Thank you for these corrections, they will be done in the final manuscript.

References

Amato, P., Demeer, F., Melaouhi, A., Fontanella, S., Martin-Biesse, A. S., Sancelme, M., Laj, P. and Delort, A. M.: A fate for organic acids, formaldehyde and methanol in cloud water: their biotransformation by micro-organisms, Atmos. Chem. Phys., 7(15), 4159–4169, doi:10.5194/acp-7-4159-2007, 2007.

Amato, P., Joly, M., Besaury, L., Oudart, A., Taib, N., Moné, A. I., Deguillaume, L., Delort, A. M. and Debroas, D.: Active microorganisms thrive among extremely diverse communities in cloud water, PLoS One, doi:10.1371/journal.pone.0182869, 2017.

Arakaki, T., Anastasio, C., Kuroki, Y., Nakajima, H., Okada, K., Kotani, Y., Handa, D., Azechi, S., Kimura, T., Tsuhako, A. and Miyagi, Y.: A general scavenging rate constant for reaction of hydroxyl radical with organic carbon in atmospheric waters, Environ. Sci. Technol., 47(15), 8196–8203, doi:10.1021/es401927b, 2013.

Besaury, L., Amato, P., Wirgot, N., Sancelme, M. and Delort, A. M.: Draft genome sequence of Pseudomonas graminis PDD-13b-3, a model strain isolated from cloud water, Genome Announc., doi:10.1128/genomeA.00464-17, 2017a.

Besaury, L., Amato, P., Sancelme, M. and Delort, A. M.: Draft genome sequence of Pseudomonas syringae PDD-32b-74, a model strain for ice-nucleation studies in the atmosphere, Genome Announc., doi:10.1128/genomeA.00742-17, 2017b.

Bianco, A., Passananti, M., Perroux, H., Voyard, G., Mouchel-Vallon, C., Chaumerliac, N., Mailhot, G., Deguillaume, L. and Brigante, M.: A better understanding of hydroxyl radical photochemical sources in cloud waters collected at the puy de Dôme station – experimental versus modelled formation rates, Atmos. Chem. Phys., 15(16), 9191–9202, doi:10.5194/acp-15-9191-2015, 2015.

Bianco, A., Voyard, G., Deguillaume, L., Mailhot, G. and Brigante, M.: Improving the characterization

of dissolved organic carbon in cloud water: Amino acids and their impact on the oxidant capacity, , 6, 37420, doi:10.1038/srep37420 https://www.nature.com/articles/srep37420#supplementary-information, 2016a.

Bianco, A., Passananti, M., Deguillaume, L., Mailhot, G. and Brigante, M.: Tryptophan and tryptophan-like substances in cloud water: Occurrence and photochemical fate, Atmos. Environ., 137, 53–61, doi:10.1016/j.atmosenv.2016.04.034, 2016b.

Deguillaume, L., Charbouillot, T., Joly, M., Vaïtilingom, M., Parazols, M., Marinoni, A., Amato, P., Delort, A. M., Vinatier, V., Flossmann, A., Chaumerliac, N., Pichon, J. M., Houdier, S., Laj, P., Sellegri, K., Colomb, A., Brigante, M. and Mailhot, G.: Classification of clouds sampled at the puy de Dôme (France) based on 10 yr of monitoring of their physicochemical properties, Atmos. Chem. Phys., 14(3), 1485–1506, doi:10.5194/acp-14-1485-2014, 2014.

Ervens, B., George, C., Williams, J. E., Buxton, G. V, Salmon, G. A., Bydder, M., Wilkinson, F., Dentener, F., Mirabel, P., Wolke, R. and Herrmann, H.: CAPRAM2.4 (MODAC mechanism): An extended and condensed tropospheric aqueous phase mechanism and its application, J. Geophys. Res., 108(D14), 4426, doi:doi: 10.1029/2002JD002202, 2003.

Ervens, B., Carlton, A. G., Turpin, B. J., Altieri, K. E., Kreidenweis, S. M. and Feingold, G.: Secondary organic aerosol yields from cloud-processing of isoprene oxidation products, Geophys. Res. Lett., 35(2), L02816, doi:10.1029/2007gl031828, 2008.

Faust, B. C. and Allen, J. M.: Aqueous-phase photochemical sources of peroxyl radicals and singlet molecular oxygen in clouds and fog, J. Geophys. Res. Atmos., 97(D12), 12913–12926, doi:10.1029/92JD00843, 1992.

Helin, A., Sietiö, O.-M., Heinonsalo, J., Bäck, J., Riekkola, M.-L. and Parshintsev, J.: Characterization of free amino acids, bacteria and fungi in size-segregated atmospheric aerosols in boreal forest: seasonal patterns, abundances and size distributions, Atmos. Chem. Phys., 17(21), 13089–13101, doi:10.5194/acp-17-13089-2017, 2017.

Herrmann, H., Hoffmann, D., Schaefer, T., Bräuer, P. and Tilgner, A.: Tropospheric Aqueous-Phase Free-Radical Chemistry: Radical Sources, Spectra, Reaction Kinetics and Prediction Tools, ChemPhysChem, 11(18), 3796–3822, doi:10.1002/cphc.201000533, 2010.

Hu, W., Niu, H., Murata, K., Wu, Z., Hu, M., Kojima, T. and Zhang, D.: Bacteria in atmospheric waters: Detection, characteristics and implications, Atmos. Environ., 179, 201–221, doi:10.1016/j.atmosenv.2018.02.026, 2018.

Kaur, R. and Anastasio, C.: Light absorption and the photoformation of hydroxyl radical and singlet oxygen in fog waters, Atmos. Environ., 164, 387–397, doi:https://doi.org/10.1016/j.atmosenv.2017.06.006, 2017.

Khaled, A., Zhang, M., Amato, P., Delort, A.-M. and Ervens, B.: Biodegradation by bacteria in clouds: An underestimated sink for some organics in the atmospheric multiphase system, Atmos. Chem. Phys. Discuss., 2020, 1–32, doi:10.5194/acp-2020-778, 2020.

Lallement, A., Besaury, L., Eyheraguibel, B., Amato, P., Sancelme, M., Mailhot, G. and Delort, A. M.: Draft Genome Sequence of Rhodococcus enclensis 23b-28, a Model Strain Isolated from Cloud Water, Genome Announc., 5(43), e01199-17, doi:10.1128/genomeA.01199-17, 2017.

Mopper, K. and Zika, R. G.: Free amino acids in marine rains: evidence for oxidation and potential role in nitrogen cycling, Nature, 325(6101), 246–249, doi:10.1038/325246a0, 1987.

Sattler, B., Puxbaum, H. and Psenner, B.: Bacterial growth in supercooled cloud droplets, Geophys. Res. Lett., 28(2), 239–242, doi:10.1029/2000GL011684, 2001.

Seinfeld, J. H. and Pandis, S. N.: Atmospheric Chemistry and Physics - From air pollution to climate change, 2nd ed., edited by I. John Wiley and Sons, John Wiley & Sons, Inc., Hoboken, New Jersey., 2006.

Tilgner, A., Bräuer, P., Wolke, R. and Herrmann, H.: Modelling multiphase chemistry in deliquescent aerosols and clouds using CAPRAM3.0i, J. Atmos. Chem., 70(3), 221–256, doi:10.1007/s10874-013-9267-4, 2013.

Vaïtilingom, M., Amato, P., Sancelme, M., Laj, P., Leriche, M. and Delort, A.-M.: Contribution of Microbial Activity to Carbon Chemistry in Clouds, Appl. Environ. Microbiol., 76(1), 23–29, doi:10.1128/AEM.01127-09, 2010.

Vaïtilingom, M., Charbouillot, T., Deguillaume, L., Maisonobe, R., Parazols, M., Amato, P., Sancelme, M. and Delort, A. M.: Atmospheric chemistry of carboxylic acids: microbial implication versus photochemistry, Atmos. Chem. Phys., 11(16), 8721–8733, doi:10.5194/acp-11-8721-2011, 2011.

Vaïtilingom, M., Attard, E., Gaiani, N., Sancelme, M., Deguillaume, L., Flossmann, A. I., Amato, P. and Delort, A.-M.: Long-term features of cloud microbiology at the puy de Dôme (France), Atmos. Environ., 56(0), 88–100, doi:http://dx.doi.org/10.1016/j.atmosenv.2012.03.072, 2012.

Vaïtilingom, M., Deguillaume, L., Vinatier, V., Sancelme, M., Amato, P., Chaumerliac, N. and Delort, A.-M.: Potential impact of microbial activity on the oxidant capacity and organic carbon budget in clouds, Proc. Natl. Acad. Sci., 110(2), 559–564, doi:10.1073/pnas.1205743110, 2013.

Xu, Y., Wu, D., Xiao, H. and Zhou, J.: Dissolved hydrolyzed amino acids in precipitation in suburban Guiyang, southwestern China: Seasonal variations and potential atmospheric processes, Atmos. Environ., 211, 247–255, doi:https://doi.org/10.1016/j.atmosenv.2019.05.011, 2019.

Yan, G., Kim, G., Kim, J., Jeong, Y.-S. and Kim, Y. Il: Dissolved total hydrolyzable enantiomeric amino acids in precipitation: Implications on bacterial contributions to atmospheric organic matter, Geochim. Cosmochim. Acta, 153, 1–14, doi:10.1016/j.gca.2015.01.005, 2015.

Zhang, M., Khaled, A., Amato, P., Delort, A.-M. and Ervens, B.: The effect of biological particles and their ageing processes on aerosol radiative properties: Model sensitivity studies, Atmos. Chem. Phys. Discuss., 2020, 1–40, doi:10.5194/acp-2020-781, 2020.